



# Dynamic response of an Arctic epishelf lake to seasonal and long-term forcing: implications for ice shelf thickness

Andrew K. Hamilton[1,2], Bernard E. Laval[1], Derek R. Mueller[2], Warwick F. Vincent[3], and Luke Copland[4]

[1]Department of Civil Engineering, University of British Columbia, Vancouver, British Columbia, Canada
[2]Geography and Environmental Studies, Carleton University, Ottawa, Ontario, Canada
[3]Department of Biology and Centre for Northern Studies (CEN), Université Laval, Quebec City, Quebec, Canada
[4]Department of Geography, Environment, and Geomatics, University of Ottawa, Ottawa, Ontario, Canada

*Correspondence to:* A. K. Hamilton (andrew@madzu.com)

**Abstract.**

Changes in the depth of the freshwater-seawater interface in epishelf lakes have been used to infer long-term changes in the thickness of ice shelves, however, little is known about the dynamics of epishelf lakes and what other factors may influence their depth. Continuous observations collected between 2011 and 2014 in the Milne Fiord epishelf lake, in the Canadian Arctic, showed that the depth of the halocline varied seasonally by up to 3.3 m, which was comparable to interannual variability. The seasonal depth variation was controlled by the magnitude of surface meltwater inflow and the hydraulics of the inferred outflow pathway, a narrow basal channel in the Milne Ice Shelf. When seasonal variation and an episodic mixing of the halocline were accounted for, long-term records of depth indicated there was no significant change in thickness of ice along the basal channel from 1983 to 2004, followed by a period of steady thinning at 0.50 m a$^{-1}$ between 2004 and 2011. Rapid thinning at 1.15 m a$^{-1}$ then occurred from 2011 to 2014, corresponded to a period of warming regional air temperatures. Continued warming is expected to lead to the breakup of the ice shelf and the imminent loss of the last known epishelf lake in the Arctic.

## 1 Introduction

Polar aquatic ecosystems that depend on ice for their physical containment, surface covering, or as a source of freshwater, are highly sensitive to climate conditions (White et al., 2007; Prowse et al., 2011; Wrona et al., 2016). Where ice dams freshwater that floats directly on seawater, such as for stamuhki lakes (ephemeral coastal lakes that form where river input is dammed behind sea ice ridges; Galand et al., 2008) or epishelf lakes (perennial lakes that form behind floating ice shelves), mass loss from the ice dam due to a shift in climate can lead to a thinning of the freshwater layer. This is particularly notable for epishelf lakes, which are known to persist for decades (Smith et al., 2006; Veillette et al., 2008) or even millennia (Doran et al., 2000; Antoniades et al., 2011) behind stable ice shelves. However, recent thinning and collapse of ice shelves in the Arctic due to climate warming (Vincent et al., 2001; Copland et al., 2007; Mueller et al., 2003; England et al., 2008; White et al., 2015a; Mueller et al., 2017a) has resulted in physical changes of epishelf lakes in the region, including the complete loss of several lakes and a substantial reduction in the depth of the freshwater layer of the few that remained (Mueller et al., 2003; Veillette et al., 2008; White et al., 2015b; Mueller et al., 2017b). Given their sensitivity to the state of the impounding ice shelf, observing



changes to the water column structure of epishelf lakes can potentially provide a relatively simple way to monitor changes in the integrity of the impounding ice shelf (Mueller et al., 2003; Veillette et al., 2008).

Epishelf lakes form in ice-free areas adjacent to floating ice shelves and maintain a hydraulic connection to the sea (Gibson and Andersen, 2002). Epishelf lakes are numerous in Antarctica and are distributed around the margins of the continental ice sheet (Heywood, 1977; Gibson and Andersen, 2002; Laybourn-Parry et al., 2006; Smith et al., 2006) and were once relatively numerous along the northern coast of Ellesmere Island, in the Canadian Arctic (Vincent et al., 2001; Veillette et al., 2008). Similar systems may exist elsewhere in the Arctic where ice shelves form, including Greenland, Franz Josef Land, and Severnaya Zemlya, but to date, no epishelf lakes have been reported from these areas. The physical properties of epishelf lakes vary depending largely on the connection to the ocean below the ice shelf. At one extreme are those where the freshwater layer floats directly on seawater, such as Ablation Lake, West Antarctica (Heywood, 1977), Beaver Lake, East Antarctica (Wand et al., 2011), and Disraeli Fiord in the Canadian Arctic (Vincent et al., 2001), while at the other extreme are epishelf lakes that are entirely fresh but have a restricted hydraulic connection to the ocean below the ice shelf (e.g., Schirmacher Oasis, East Antarctica; Bormann and Fritzsche, 1995), with lakes of varying degrees of stratification in between (e.g., southern Bunger Hills, East Antarctica; Gibson and Andersen, 2002).

For epishelf lakes where freshwater floats directly on seawater, the focus of this paper, snow and ice meltwater from the surrounding catchment is thought to accumulate until the thickness of the freshwater layer is equal to the minimum draft of the ice shelf (Hattersley-Smith, 1973; Jeffries and Krouse, 1984; Gibson and Andersen, 2002; Mueller et al., 2003). Excess freshwater inflow is then assumed to be exported below the base of the ice shelf to the open ocean. Where low tidal action and perennial ice cover limit mixing between the fresh and saltwater layers, the halocline can be unusually sharp (with reported vertical salinity gradients of $\sim$40 ppt m$^{-1}$; Heywood, 1977) and stable. The strong vertical physical and chemical stratification creates a rare ecosystem where fresh, brackish, and salt water biota can exist within a single water column (Laybourn-Parry et al., 2006; Veillette et al., 2011a). Thus changes to the physical structure of the water column can affect ecosystem functioning (Thaler et al., 2017).

Measuring changes in the depth of the halocline separating freshwater and seawater could provide a relatively straightforward means to infer changes in ice shelf thickness (Vincent et al., 2001; Mueller et al., 2003; Veillette et al., 2008). This has been undertaken extensively along the northern coast of Ellesmere Island, in the Canadian Arctic, where the climate has warmed at twice the global average (IPCC, 2013). For example, water column profiles collected in Disraeli Fiord behind the Ward Hunt Ice Shelf (WHIS), Ellesmere Island, showed that its epishelf thinned from a maximum depth of 63 m in 1954 to 33 m in 1999 (Veillette et al., 2008), suggesting a long-term thinning of the WHIS prior to its fracturing between 2001 and 2002, which resulted in the complete drainage of the epishelf lake (Mueller et al., 2003).

Questions remain, however, as to what extent changes in the depth and gradient of an epishelf lake halocline are related to changes in the ice shelf, and what other factors may be important (see Fig. 1 for a schematic of some of the processes discussed below). Vincent et al. (2001) suggested that the long-term thinning of the Disraeli Fiord epishelf lake could have been the result of preferential drainage of freshwater via a localized conduit at the base of the WHIS, not necessarily representative of changes in the mean thickness of the ice shelf. Veillette et al. (2008) suggested that observed interannual increases in the halocline depth





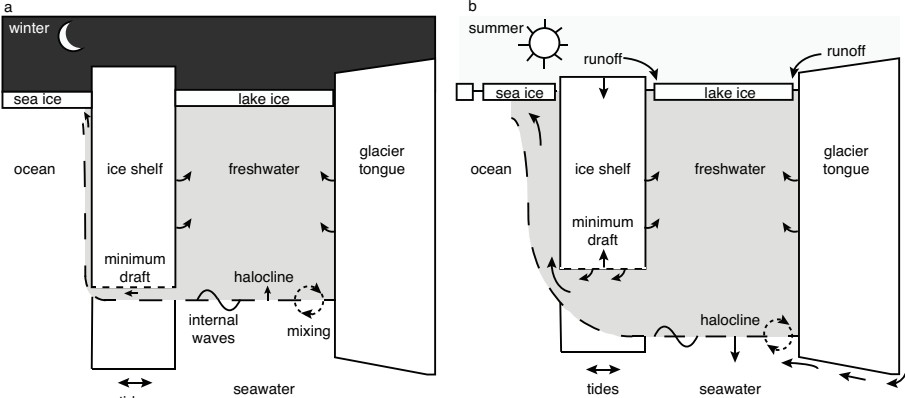

**Figure 1.** Schematic of the Milne Fiord epishelf lake showing processes that could influence the depth of the halocline in (a) winter and (b) summer. Potential influences include tidal flow, internal waves, mixing, deepening of the freshwater layer due to seasonal inflow, shoaling of the freshwater layer due to outflow beneath the ice shelf, changes the mean thickness of the ice shelf or melting along the outflow pathway changing the minimum draft of the ice shelf. Arrows indicating submarine melting of the walls of the ice shelf and glacier tongue and meltwater discharged at the base of the glacier tongue are included for completeness, although neither process will alter the depth of the epishelf lake. The ice walls are in hydrostatic equilibrium so their melting will add volume to the the freshwater layer but not change the depth. Meltwater runoff discharged at the bed of the glacier will not penetrate to the freshwater layer because the vertical ascent of the buoyant plume will be limited as its density increases through entrainment of seawater, until it spreads out below the epishelf lake halocline as a subsurface plume (Hamilton, 2016).

of some Ellesmere Island epishelf lakes were not indicative of a thickening of the ice shelves, but were more likely related to short-term vertical oscillations of the halocline due to tidal cycles, internal waves, or fjord circulation. Similarly, Smith et al. (2006) suggested the shoaling and weakening of the halocline in Ablation Lake, Antarctica, was not related to thinning of the George IV Ice Shelf, but was more likely to reflect seasonal changes in the supply of fresh meltwater, or variation due to changes in tides or the hydraulic connection to the sea below the ice shelf (e.g., Galton-Fenzi et al., 2012). Understanding of epishelf lake systems is, however, hampered by the fact that long-term records are based almost entirely on sparse conductivity-temperature-depth (CTD) profiles collected over intervals of years, or even decades, with little knowledge of the seasonal dynamics or spatial heterogeneity of these systems. Many studies have stressed the need for continuous monitoring to better understand the dynamics of epishelf lakes and their relationship to ice shelf thickness (Vincent et al., 2001; Smith et al., 2006; Veillette et al., 2008).

In this study we aimed to identify factors controlling changes in the water column structure and the depth of the halocline of last known epishelf lake in the Arctic, in Milne Fiord, Ellesmere Island, and to evaluate how these variations were related to changes in the spatial extent of the lake and to the state of the Milne Ice Shelf. We compiled archived CTD data collected in Milne Fiord since 1983, conducted extensive new CTD profiling between 2009 and 2014, and analyzed satellite data to monitor the long-term changes in the vertical structure and spatial extent of the lake. We deployed a mooring from 2011 to





2014 to continuously record changes in epishelf lake properties to investigate short-term variability, on timescales from tidal to seasonal, and determine the factors driving changes in the halocline. The complete epishelf lake halocline depth records are used to infer changes in the thickness and state of the Milne Ice Shelf and we discuss the implications of our findings on the interpretation of records from other ice shelf–epishelf lake systems.

## 1.1 Study site and background

Milne Fiord (82°35' N, 80°35' W) is 436 m deep and lies on the northern coast of Ellesmere Island adjacent to the Arctic Ocean (Fig. 2). The Milne Fiord epishelf lake (MEL) is dammed by the Milne Ice Shelf (MIS), which spans 18 km across the mouth of the fjord. At the head of the fjord the Milne Glacier terminates in a 16 km long glacier tongue that forms the landward margin of the lake. The perennial freshwater ice cover of the MEL, which is approximately 1 m thick, extends throughout the inner fjord and appears as an area of high backscatter (light) in Synthetic Aperture Radar (SAR) imagery, in contrast to the low backscatter (dark) of the MIS and the glacier tongue (Fig. 2a). From SAR imagery, Mortimer (2011) estimated the area of the epishelf lake as 52.5 km$^2$ in 2009, appearing to consist of a 6 km wide main basin between the inner edge of the MIS and the terminus of the Milne Glacier tongue, and two narrow arms extending 16 km along the sides of the glacier tongue to the Milne Glacier grounding line, although the actual spatial extent of the freshwater layer had not been confirmed with field observations.

The MEL was first discovered in 1983, when Jeffries (1985) collected water samples through 3.19 m of surface ice that revealed a 17.5 m deep freshwater layer separated from seawater by a sharp halocline only a few metres thick. The lake was not sampled again until 2004, when CTD profiles showed the freshwater layer had deepened to 18.3 m; by 2009 it had thinned to 14.3 m, which was assumed to correspond to the minimum draft of the MIS (Veillette et al., 2011b).

Surveys of the MIS have shown that its ice thickness varies from 94 m to 8 m (Fig. 3; Mortimer et al., 2012; Hamilton, 2016). The topology of the ice shelf is quite variable owing to its complex origins, and has been divided into three regions based on surface morphology and ice characteristics (Jeffries, 1986). The previously defined Inner Unit, which once abutted the Milne Glacier tongue, has been replaced with epishelf lake ice since the 1990s (Mortimer et al., 2012). The Central Unit currently forms the landward (southern) edge of the ice shelf, and has an erratic surface morphology and highly variable thickness owing to substantial past input from tributary glaciers, and is apparent as an region of low backscatter (dark) in SAR imagery (Fig. 2a). The Outer Unit has a more uniform thickness (mean thickness ∼50 m) owing to its origin from marine ice accretion and snow accumulation, and is distinguishable by a series of surface ridges and troughs running parallel to the coast, and relatively high backscatter in SAR imagery (Fig. 2a). The Outer Unit, however, is bisected by two re-healed fractures that have existed since at least 1950 (Hattersley-Smith et al., 1969). Ice thickness surveys indicate that the only ice thin enough to provide an outlet to the epishelf lake lies along these re-healed fractures (Fig. 3), although the drainage pathway of the lake has not been confirmed by observations. The MIS is in a state of negative mass balance (Mortimer et al., 2012) and is not expected to regenerate in the current climate (Copland et al., 2007; Veillette et al., 2011a; White et al., 2015a).





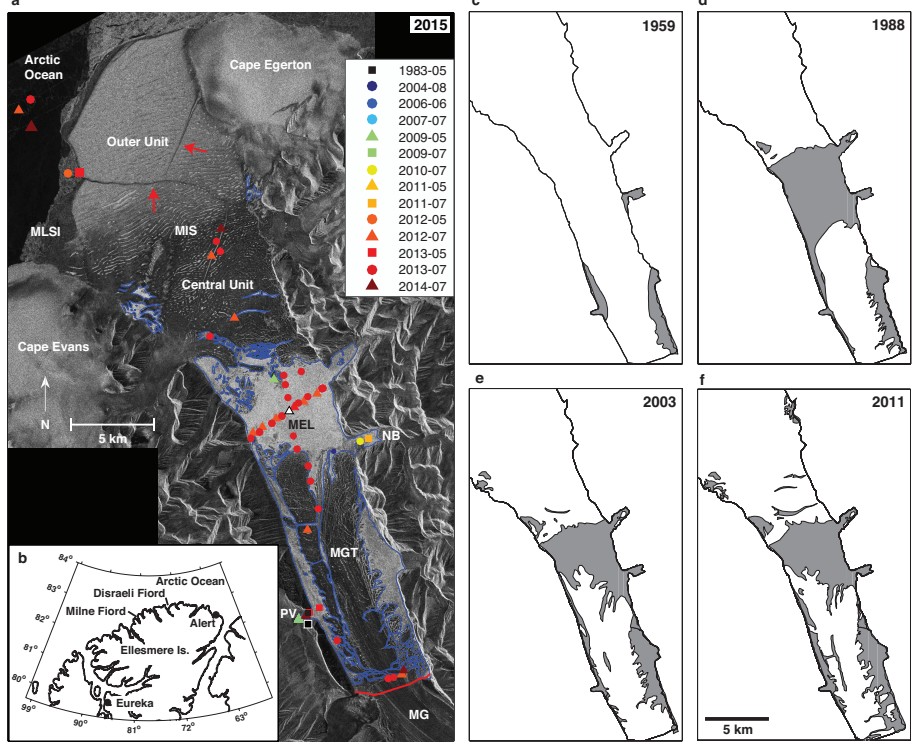

**Figure 2.** Map of Milne Fiord study area. (a) RADARSAT-2 image mosaic of Milne Fiord showing the extent of the Milne Fiord epishelf lake (MEL) in 2015, corresponding to the region of high backscatter (light gray) outlined in blue. The locations of CTD profiles collected during each field campaign are shown. The Central and Outer Units of the Milne Ice Shelf (MIS), as well as two re-healed fractures (red arrows), are indicated, as well as the Milne Glacier (MG), grounding line (red line), Milne Glacier tongue (MGT), multiyear landfast sea ice (MLSI), the met station (white and black square), mooring (white and black triangle), and two small inlets unoffically named Purple Valley Bay (PV) and Neige Bay (NB). (b) Regional map of Ellesmere Island, Canada. The sequence of four panels on the right show the increase in area of the MEL (gray) estimated from aerial and satellite imagery from (c) 1959, (d) 1988, (e) 2003, and (f) 2011, based on data from Mueller et al. (2017a). The coastline of Milne Fiord is outlined in black. White areas inside the coastline are glacier or ice shelf.

## 2 Methods

### 2.1 Area and volume

To track changes in the extent of the MEL, its area was estimated from optical imagery acquired by various aerial and satellite

platforms from 1959 to 1988, and from 1992 onward from SAR imagery (Mueller et al., 2017b). The epishelf lake ice was discriminated from other surrounding ice types, including ice shelf, glacier ice and marine ice, in optical imagery by its lack of surface topography, and in winter SAR imagery by its high backscatter signal (>-6 dB), produced by its lack of surface topography and freshwater underneath perennial lake ice (thin ice directly underlain by salt water has a darker return; Veillette





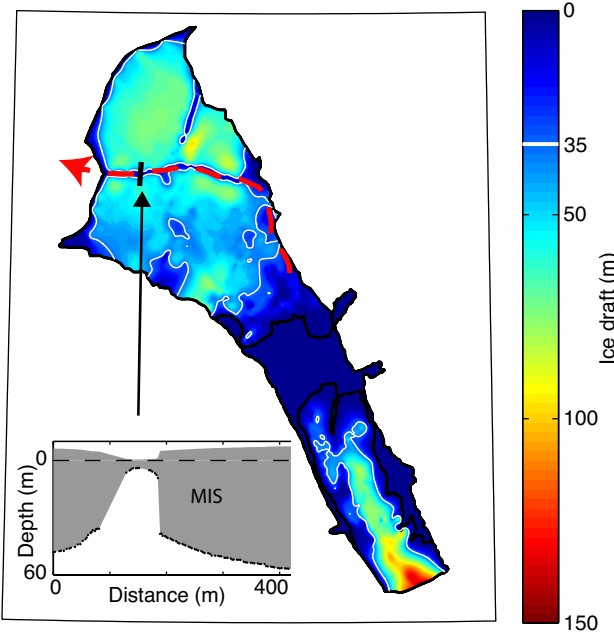

**Figure 3.** Map of the draft of ice in Milne Fiord indicating the likely drainage pathway of the MEL along a re-healed fracture in the MIS. The modelled ice draft is that presented in Hamilton (2016) from ground-based, aerial, and satellite ice thickness and surface elevation data assuming the ice is in hydrostatic equilibrium. The 35 m contour is highlighted to show the outflow restriction the MIS imposes across the mouth of the fjord for water above this level. The inset shows ice thickness across the basal channel from a 400 m ice-penetrating radar transect that crossed the re-healed fracture in July 2013 (Hamilton, 2016).

et al., 2008; White et al., 2015a). The epishelf lake was manually digitized in ArcGIS 10.2.2 at an image scale of ∼1:20,000,
with a pixel size of 6.5 m, and the area of the resultant polygons were calculated. Non-contiguous regions of lake ice in fractures in the MIS and between calved pieces of the MIS and Milne Glacier tongue were included in area estimates. Lake volume was estimated from area and depth, assuming a spatially uniform depth (see Section 3.3) and vertical shores. The volume estimated included the volume of lake surface ice. The depth used for volume calculations was estimated from hydrographic profiles (see Section 2.3) collected closest to the date of image acquisition (see Table 1).

**2.2 Hydrography**

Long-term changes in the structure of the epishelf lake were monitored using archived and newly acquired water column profiles. The lake was first profiled in 1983 using reversing thermometers and a Radiometer CDM80 Conductivity Meter (Jeffries, 1985), and we present this data here. Near-annual CTD profiling commenced in 2004, with a directed and intensive sampling program from 2011 to 2014. CTD profiles were collected through drilled holes or natural leads in the ice, including
fractures through the ice shelf and glacier tongue, accessed by foot, snowmobile, or helicopter. Opportunistic profiles were collected in August 2004, June 2006, and July 2007 using a 1 Hz RBR XR-420 CTD. Subsequent profiles were collected in





May and July of 2009, July 2010, May and July of 2011, 2012, and 2013, and July 2014 with a 6 Hz RBR XR-620 CTD. Profiles from May 2011 were collected using a 4 Hz Seabird SBE19+ CTD, and in July 2011 using a 1 Hz Hydrolab HLX. The 2004 CTD profile was collected just north of the Milne Glacier tongue terminus, and from 2006 to 2010 profiles were collected in either Purple Valley Bay or Neige Bay (unofficial names), two shallow inlets on the west and east sides of the fjord, respectively (Fig 2a). From 2011 onward multiple profiles were collected during each field campaign throughout the main fjord. Full depth profiles were usually collected to the bottom of the fjord, but here we focus on the upper 25 m of the water column. CTDs were

calibrated once every 2 years after 2011, prior to this the CTDs were not regularly calibrated so we interpret data prior to 2011 with this caveat on absolute accuracy. Profiles collected between 2004 and 2009 were previously published in Veillette et al. (2011b), although we have reprocessed all these data from raw conductivity and temperature measurements (where available) for consistency.

CTD data were processed in Matlab following a procedure that included: correction for atmospheric pressure, application of

a 3 point low-pass filter in time to the raw pressure, temperature, and conductivity; alignment of conductivity and temperature with respect to pressure; a thermal cell mass correction (for the SBE19+ CTD data only); and loop editing (removal of pressure reversals), and bin averaged to 0.2 m intervals. Derived variables were calculated using the International Thermodynamic Equation of Seawater 2010 (TEOS-10) Gibbs Seawater Oceanographic Matlab Toolbox (www.TEOS-10.org), with Conservative Temperature ($\Theta$) and Absolute Salinity ($S_A$) reported. For freshwater lakes, the determination of salinity from measured

temperature, conductivity, and pressure is dependent on the chemical composition of the water (Pawlowicz, 2008), data which we lack. However, the conductivity of two inflowing meltwater streams measured in 2012 and 2013 was low (<0.06 mS cm$^{-1}$), suggesting that the source of ions to the lake (with conductivity generally >0.15 mS cm$^{-1}$) was the underlying seawater and the use of TEOS-10, which assumes seawater composition, was appropriate.

## 2.3 Lake depth from CTD profiles

In order to compare changes in the thickness of the freshwater layer over time it was necessary to define the bottom of the epishelf lake, which was actually a continuum from freshwater to seawater. Previous studies have used the depth of the 3 ppt isohaline (Mueller et al., 2003; Veillette et al., 2008), or the depth of the halocline (which the authors qualitatively define as the zone of abrupt salinity change between freshwater and sea water; Veillette et al., 2011b). We formalized the definition of Veillette et al. (2011b) by defining the bottom of the lake ($D_{EL}$) as the depth of the stratification maximum as defined by the

Brunt–Väisälä frequency:

$$D_{EL} = z(N^2_{max}), \tag{1}$$

where $z$ is depth (positive downward) and $N^2$ is the Brunt–Väisälä frequency:

$$N^2 = \frac{g}{\rho}\frac{\partial \rho}{\partial z}, \tag{2}$$

where $g$ is gravitational acceleration, and $\rho$ is density of the water. Profiles of $N^2$ were averaged using a 12-point depth window

before calculating the maximum. The advantage of this method is that the epishelf lake depth calculation is clearly defined,





quantitative, and not dependent on the absolute salt content of the epishelf lake (which would affect the 3 ppt method). Due to the bottle sampling method used by Jeffries (1985) we could not accurately calculate $N^2$ for that profile, so we defined the bottom of the lake as the depth of the sample collected nearest the apparent stratification maximum. The depth of the lake as measured by a series of 18 CTD profiles collected at a single location over 24 hours in May 2009 varied by $\pm$ 0.2 m, so this

was considered the depth measurement uncertainty from CTD profiles.

## 2.4 Current velocities

To understand the probability of shear induced vertical mixing of the halocline velocities of the upper water column were measured using an ice-anchored, downward-looking 300 kHz RDI acoustic Doppler current profiler (ADCP) at the mooring site in the centre of the MEL (Fig. 2a) over 4 days in May 2011, 7 days in July 2012 and 10 days in July 2013. The ADCP

sampled at 2 min intervals, 2 m bins, with 150 pings per ensemble, and the data were processed in Matlab.

## 2.5 Tidal height

Changes in water level were recorded using a bottom anchored RBR XR-620 CTD (accuracy is $\pm$ 0.37 dbar, drift 0.74 dbar $a^{-1}$) deployed at 355 m depth at the mooring site (Fig. 2a) from May 2011 to July 2012 sampling at 2 minute intervals.

## 2.6 Meteorological time series

Meteorological conditions were recorded by a HOBO automated weather station (AWS) located at 10 m elevation in Purple Valley Bay (Fig. 2 a). Only air temperature (at 1 m and 2 m above ground) and shortwave solar radiation are reported here. Cumulative positive-degree days (PDDs), the daily integrated air temperatures above 0°C, were calculated to provide a direct proxy for summer surface melting (Hock, 2003), and thus inflow to the lake. Prior to the AWS installation in 2009, we estimated summer air temperatures and PDDs in Milne Fiord from air temperature records at Eureka, Nunavut, 280 km to the south

(www.ec.gc.ca). Linear regression showed that daily mean air temperature in Milne Fiord $T_{Milne}$, were correlated ($R^2 = 0.92$) to daily mean air temperature in Eureka by $T_{Milne} = 0.87 * T_{Eureka} - 3.26$. $T_{Milne}$.

## 2.7 Mooring time series

Milne Fiord water properties were recorded continuously from May 2011 to July 2014 from a mooring deployed in the centre of the epishelf lake (Fig. 2a). The mooring was anchored to the epishelf lake ice and suspended down the water column.

The mooring consisted of 20 RBR TR1050/60 temperature sensors, 2 RBR XR420-freshwater conductivity-temperature (CT) sensors, two Seabird SBE37 CTs, and one RBR XR620 CTD from May 2011 to July 2012, then was reduced to seven TR1060s, one XR420 CT, and one XR620 CTD for the remainder of the study. Calibrated accuracy for all sensors is $\pm 0.002$°C, $\pm$ 0.003 mS cm$^{-1}$, and $\pm$ 0.37 dbar, and nominal drift is $\pm 0.002$°C a$^{-1}$, $\pm$ 0.012 mS cm$^{-1}$a$^{-1}$, and $\pm$ 0.7 dbar a$^{-1}$. Some time series records were truncated for various reasons, including salinity going beyond the maximum sensor range (3 PSU) of the

freshwater instruments (XR420s), or instrument malfunction. The mooring was serviced once or twice per year and instruments





were repositioned to track the halocline. Initially, the instruments were spaced every metre from the surface to 20 m depth, with increasing depth intervals below 25 m. Although the mooring instruments extended to the seabed at 355 m, in this paper we focus on the top 25 m of the water column. Instruments sampled at 30 to 120 second intervals. CTD profiles collected during mooring deployment and recovery were used to correct for instrument drift, which was within manufacturer specifications.

## 2.8 Lake depth from mooring timeseries

Seasonal changes in the depth of the epishelf lake were estimated from changes in salinity recorded by the conductivity sensor initially positioned within the halocline (at 13 m depth from May 2011 to May 2012). Salinity changes at a fixed depth over time were assumed to be driven by a vertical displacement of an otherwise unchanging initial vertical salinity gradient (obtained during CTD profiling). For example, an increase (decrease) in salinity was assumed to be caused by an upward (downward) displacement of the halocline. This method neglected other processes that could alter the salinity gradient, including horizontal advection of water masses or vertical mixing, and we discuss the impact of these in other sections.

We also estimated epishelf lake depth using temperature data from the more numerous and closely spaced thermistors on the mooring. The thermocline is coincident with the halocline because cold seawater below the halocline can circulate horizontally, and continually remove heat at the base of the lake. Lake depth was therefore estimated from the depth of the isotherm corresponding to the average temperature at the depth of the $N_{max}^2$ measured by CTD profiles collected at the beginning and end of each mooring deployment. Temperature changes at a fixed depth were assumed to be caused by a vertical displacement of an otherwise unchanging thermocline (and halocline), and other processes that could alter the vertical distribution of heat, such as heating due to solar radiation, horizontal advection, and vertical heat flux or mixing across the halocline, were neglected. While these are broad assumptions, we note that all three methods used to determine lake depth (CTD profiling, the moored conductivity sensors, and moored thermistors) showed good agreement (with one notable exception detailed in Section 3.5.4), providing confidence in the methods.

## 3 Results

### 3.1 Stratification

The most conspicuous feature of all water column profiles collected in Milne Fiord was the presence of a several meter thick relatively warm, fresh (0–3°C, ~0.2 g kg$^{-1}$) layer that was separated from cool, saline (<-1°C, ~30 g kg$^{-1}$) water by a sharp halocline and thermocline (Fig. 4). The epishelf lake was apparent in the first water samples obtained in the fjord in 1983 (Jeffries, 1985), but it clearly thinned through time. Despite changes in the depth of the lake, several distinct layers in the epishelf lake and upper water column could be identified based on salinity and temperature characteristics (Fig 2c).

In summer, a 1–2 m thick stratified layer with salinity <0.2 g kg$^{-1}$ and temperature approaching the freshwater freezing point (0°C) was present just below the surface ice–water interface. We termed this the surface melt layer given its origins from local melting of surface ice. Below the thin surface melt layer was a layer of nearly constant salinity (approximately 0.2 g





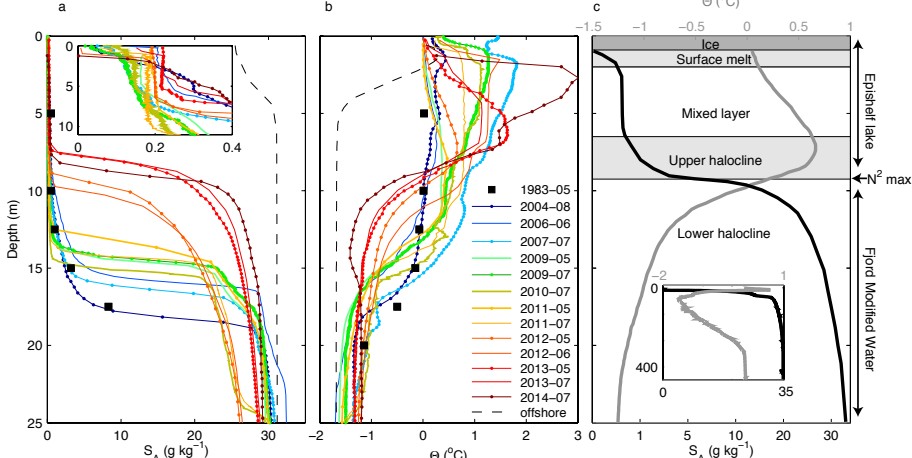

**Figure 4.** Changes in (a) salinity ($S_A$) and (b) temperature ($\Theta$) properties of the upper 25 m of Milne Fiord from all field campaigns from 1983 to 2014. A single representative profile collected at the mooring site from each field campaign is shown when multiple profiles were collected. Inset in (a) shows a zoom in of epishelf lake salinities. Dashed lines in (a) and (b) indicate a representative profile collected offshore of the MIS. (c) Idealized salinity (black line) and temperature (gray line) profiles showing the layers of the upper water column. Note the non-linear salinity scale. The epishelf lake is defined as extending from the surface to the buoyancy frequency maximum ($N^2_{max}$). The inset shows the full water column properties of the fjord to 440 m depth.

kg$^{-1}$), the mixed layer, extending from the base of the surface melt layer to the top of the halocline. The mixed layer was up to 8 m thick, with temperature between 0–2°C, however it was not present in all years, and usually only evident in summer; at other times the lake was weakly salinity stratified.

20    The strong salinity gradient below the mixed layer was divided into an upper and lower halocline. The upper halocline was the transition from the base of the mixed layer to the bottom of the lake (i.e., $z(N^2_{max})$). A subsurface temperature maximum (up to 3°C) was usually associated with the upper halocline. When the mixed layer was not present, the upper halocline extended to the base of the surface ice melt layer (if present) or to the ice–water interface. The gradient and thickness of the upper halocline varied among years, with a thicker and more gradual salinity gradient apparent prior to 2009 (e.g., in 2004 the upper halocline was 15 m thick and extended almost to the surface), while after 2009 the salinity gradient was thin and sharp (e.g., in June 2012 the upper halocline was <3 m thick). The lower halocline was defined as extending below the $z(N^2_{max})$ to the level at which properties within the fjord were equivalent to those at the same depth offshore (between 25 and 50 m). Temperatures in the lower halocline decreased rapidly with depth toward the freezing point of seawater (<-1°C). The properties of the lower halocline varied among years and were likely dependent on local fjord processes, including interactions with ice and advection of subsurface glacial meltwater runoff, so the lower halocline was also referred to as fjord-modified water (Hamilton, 2016).

    The salinity profiles showed a clear long-term thinning of the freshwater layer, from a maximum depth of 18.3 m in 2004 to a minimum of 8.0 m depth in 2013 (Fig. 4; Table 1). There was little change in the depth of the halocline between 1983 and 2004, then substantial thinning between 2004 and 2014. However, the CTD profiles indicate the magnitude and direction of change





was not constant over time. For example, the halocline increase in depth between some years (e.g., 2006–2007, 2009–2010, and 2013–2014), while an apparent abrupt thinning occurred between 2011 and 2012. These changes are discussed in more detail below.

## 3.2 Current velocities

The relatively brief ADCP deployments (<10 days) indicated a quiescent system with currents <2 cm s$^{-1}$ in the upper 25 m of the water column (not shown). The currents were weakly baroclinic, with velocities near zero in the epishelf lake above the level of the halocline, increasing to 1–2 cm s$^{-1}$ just below the halocline. The potential for velocity shear stress to generate vertical mixing in the water column was determined by calculating the gradient Richardson Number, a ratio of stratification to velocity shear:

$$Ri = \frac{N^2}{\left(\frac{\partial u}{\partial z}\right)^2} \tag{3}$$

where $u$ is horizontal velocity (m s$^{-1}$) and $z$ is depth (m; positive z down). During all three periods of observation $Ri \gg 1$ across the halocline, indicating that stabilizing buoyancy forces dominate and turbulent mixing was not expected.

## 3.3 Spatial extent

Although the depth of the halocline varied over time, the epishelf lake was present in all profiles collected landward (south) of the Outer Unit of the MIS in all years. Transects occupied over periods <24 h showed very little spatial heterogeneity in the depth of the halocline throughout the fjord. For example, $z(N_{max}^2)$ varied by only ± 0.1 m in three profiles collected by helicopter on 29 June 2012 over 23 km from the grounding line of the Milne Glacier to a fracture in the Central Unit of the MIS (dark blue lines in Fig. 5a). Similarly, $z(N_{max}^2)$ varied by only ± 0.05 m in six profiles collected across the 5.8 km width of the fjord (green lines in Fig. 5a). In addition, 21 profiles collected at multiple locations throughout the fjord in July 2013 showed $z(N_{max}^2)$ varied by <0.5 m over almost 3-weeks (Fig. 5b). Each of the profiling locations where the epishelf lake was present corresponded to a region of high backscatter in the RADARSAT-2 imagery acquired the winter prior, interpreted as freshwater lake ice, and provided verification of the remote sensing method used to map the extent of the lake.

That the epishelf lake was observed even through fractures in the Central Unit of the MIS indicated that a network of fractures must allow water exchange under the Central Unit with the main basin of the epishelf lake. The fact that the epishelf lake was not present in profiles collected beyond the seaward edge of the MIS indicated that the ice dam for the epishelf lake was likely located along the east–west running fracture in the Outer Unit of the MIS (Fig. 3). Several attempts in 2012, 2013, and 2014 to profile through the re-healed fractures in the Outer Unit, to constrain the location of the ice dam and confirm the drainage pathway of the MEL, were unsuccessful. Investigation into the location of the ice dam and the drainage pathway of the MEL is ongoing.





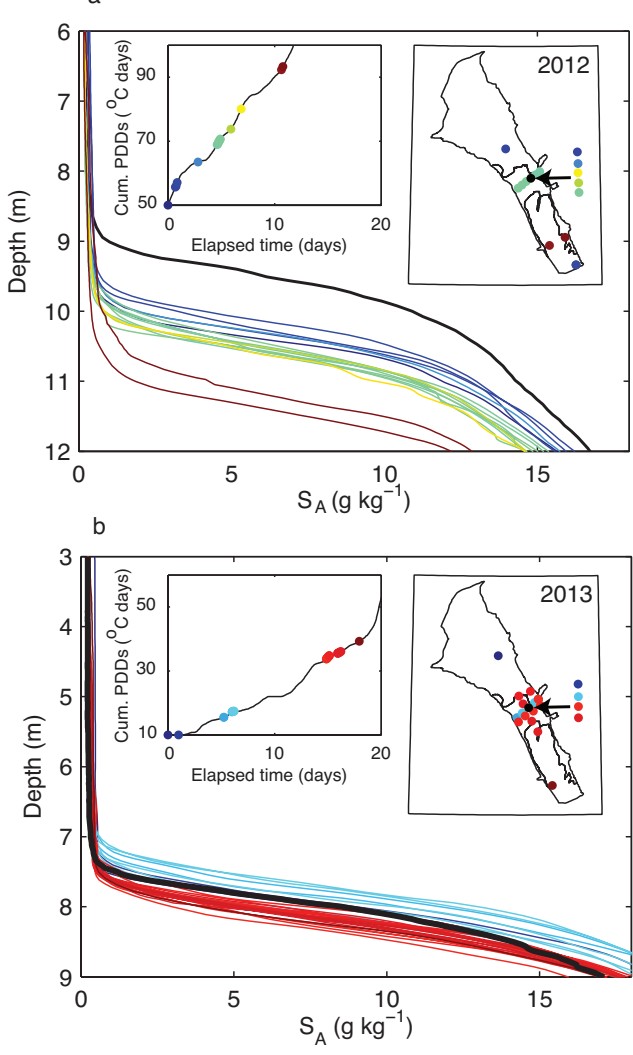

**Figure 5.** Salinity profiles showing seasonal changes in depth of the MEL halocline in a) 2012 and b) 2013. Profiles are coloured by time over the duration of each summer field campaign (10-days in June/July 2012 and 18-days in July 2013). A single profile collected in May of each year, prior to the onset of the melt season, is shown (black line). Note the different y-axes range between a) and b), although the incremental scales are consistent. In a) and b) the left inset shows the PDDs accumulated (a proxy for the volume of surface meltwater inflow) during that field campaign, with the timing of profiles indicated (coloured circles). The right inset shows the profiling locations on a map of Milne Fiord (the MEL, MIS, and Milne Glacier tongue are outlined). Multiple profiles were collected at the mooring site (black circle) during each campaign.





## 3.4 MEL area and volume

Analysis of aerial and satellite imagery indicates that the area of the lake increased substantially over time, expanding from a few small ice-marginal lakes in 1959, to close to its present form by 1988, with an area of 71.2 km$^2$ as of 27 March 2015 (Fig. 2; Table 1). The largest change in area occurred sometime between 1959 and 1988, as the epishelf lake replaced the Inner Unit of the MIS. Increases in lake area after 1992 occurred due to retreat of the southern margin of the MIS, including calving of the MIS into the fjord, the creation of small satellite lakes in fractures of the MIS, and wastage along the margins of the Milne

Glacier tongue. Some gains in area were partially offset by losses due to the advance of the terminus of the Milne Glacier tongue, which advanced 5.4 km down the fjord between 1950 and 2015.

The observed spatial uniformity of the depth of the lake (at a moment in time) allowed a straightforward determination of the volume of the lake from area and depth (Table 1). The earliest reliable estimate of the lake volume was 1.04 km$^3$ in 2006, after which the volume decreased substantially, reaching a minimum of 0.54 km$^3$ in 2013. The decrease in volume is largely

due to the decrease in the depth of the lake; the area of the lake varied by only 10% between 2006 and 2014, while the depth varied by over 50% during this period. We note that measured surface ice thickness decreased from a maximum of 3.19 m in 1983 (Jeffries, 1985) to a minimum of 0.65 m in July 2010. However, changes in surface ice thickness did not affect estimated volumes as the surface ice was in hydrostatic equilibrium.

## 3.5 Temporal variability

### 3.5.1 Tidal

Water level records revealed that the tidal range in Milne Fiord was small, with a maximum range of 0.31 m between May 2011 and May 2012. The low tidal energy available for mixing in Milne Fiord has likely been an important factor in the long-term persistence of the epishelf lake halocline.

### 3.5.2 High frequency

Salinity sensors moored in the halocline showed evidence of high frequency variations that were likely due to the passage of internal waves. Spectral analysis of the salinity time series (not shown) revealed energy peaks at diurnal and semi-diurnal tidal periods in the halocline, as well as a strong non-tidal peak at 48 min, and secondary peaks at 70 min and 5.7 h. However, at all of these periodicities salinity fluctuated by <2 g kg$^{-1}$, equivalent to a vertical displacement of the halocline of <0.2 m. The relatively low energy of the background internal wave field in Milne Fiord suggests they likely have a limited role in inducing

mixing across the halocline, and are primarily of concern as being a small source of error when determining epishelf lake depth from CTD profiles.





### 3.5.3  Seasonal

Records from the automated weather station over the period of the mooring deployment (Fig. 6) showed that although the average air temperatures were well below zero (-18.8°C from 15 May 2011 to 15 May 2014) the seasonal variation in air temperature was extreme, ranging from an hourly maximum of +20.2°C in July 2012 and to a minimum of -51.8°C in February 2013. The range in air temperature was driven, in part, by variation in solar radiation at this high latitude, which ranged from zero during the polar night (mid-October through February) to a daily maximum of ∼650 W m$^{-2}$ in late June during the period of 24 h of daylight between April and September. Summer melt conditions varied substantially among years, with cumulative positive degree days of 278, 253, 92 and 110 in 2011, 2012, 2013, 2014, respectively (note that for display purposes the meteorological record is truncated to match the mooring record in Fig. 6).

The timeseries of epishelf lake water temperature (Fig. 6e) revealed that the lake maintained a subsurface temperature maximum throughout the year, reaching a maximum of 2.5°C, 4.0°C, and 2.5°C, in 2011, 2012 and 2013, respectively. Temperatures decreased over winter but remained above freezing despite extremely low air temperatures. Below the epishelf lake there was far less seasonal variation in temperature, with the lower halocline remaining relatively cool (approximately -1°C) all year.

The most striking features of the mooring timeseries are the substantial seasonal changes in the depth of the thermocline and halocline as revealed by records of salinity (Fig. 6d) and temperature (Fig. 6e). Sensors moored at depths corresponding to the thermo/halocline (referred to hereafter as simply the halocline; between 10 m and 15 m from 2011 to 2013, and 8 to 10 m from 2013 to 2014) recorded a substantial freshening and warming from mid-June to mid-August each year, although the magnitude of change varied year to year, with little change in summer of 2013. The commencement and duration of the deepening of the halocline corresponded to the onset and duration of the surface melt season, when air temperatures were above freezing (highlighted by the gray area each year in Fig. 6). Tracking the change in depth of the isotherm that corresponded to the $z(N^2_{max})$ in May of each year indicated that the halocline deepened by 3.0 m, 3.3 m, and 1.0 m in summers 2011, 2012, and 2013, respectively.

After the epishelf lake reached its maximum depth at the cessation of the surface melt season each year, the salinity at the halocline quickly began to increase (Fig. 6d) while the temperature decreased (Fig. 6e), indicating shoaling of the halocline. We suggest this was the result of excess freshwater stored in the epishelf lake gradually draining under the ice shelf over winter (see Fig. 1). We note that the rate of shoaling each winter was not constant, as apparent by an abrupt change in depth of the thermocline that occurred in January 2012 (Fig. 6c), which is examined in more detail in the following section. The thickness of the freshwater layer reached a minimum in early June each year, with the depth of the halocline at this time consistently shallower than in June the previous year. Despite the substantial seasonal variation, the mooring records show the halocline shoaled on an interannual basis by 4.1 m between June 2011 and June 2012, 1.5 m between June 2012 and June 2013, and 1.0 m between June 2013 and June 2014.

The magnitude of deepening of the halocline during each summer appeared to be related to the intensity of the surface melt season. Changes in depth the epishelf lake were significantly correlated with the total number of PDDs accumulated over the season, having a ratio ($\Delta z/PDD$) of 0.017 m °C$^{-1}$ day$^{-1}$ (n = 226, R$^2$ = 92%, p = 0.005; Fig. 7).





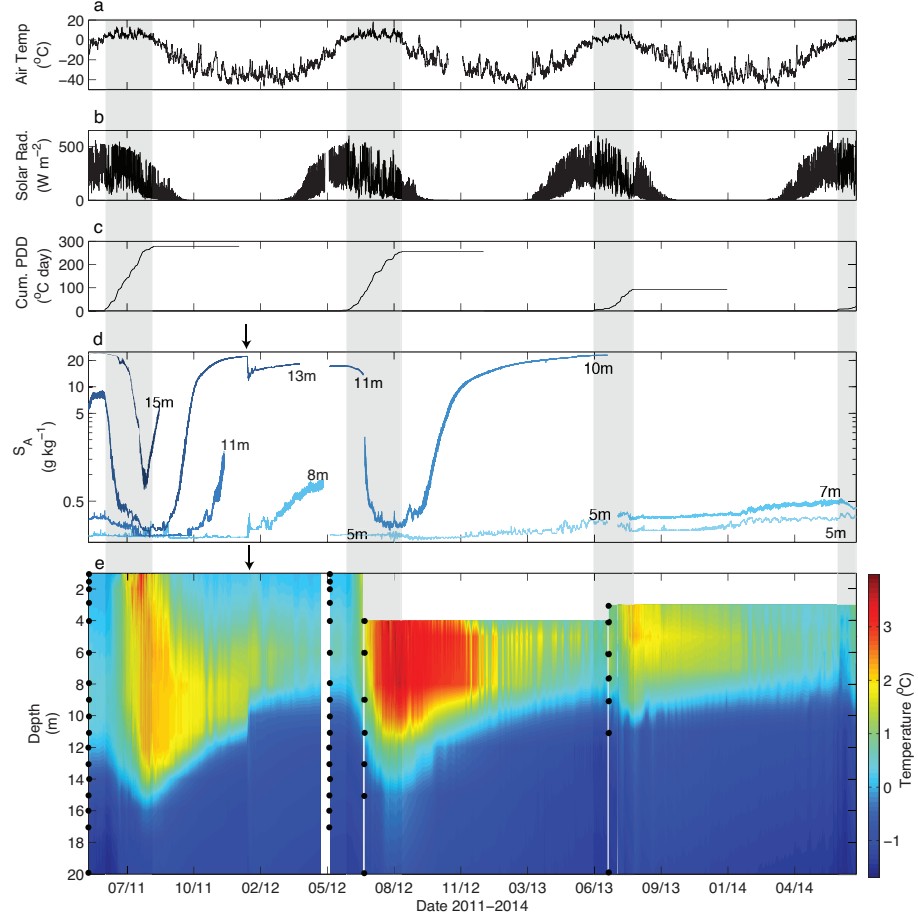

**Figure 6.** Timeseries of meteorological conditions and epishelf lake properties from May 2011 to July 2014 in Milne Fiord. (a) Air temperature. (b) Shortwave solar radiation. (c) Cumulative positive degree days (PDDs). (d) Absolute salinity ($S_A$) from instruments moored at depths between 5 m and 15 m. Note the logarithmic scale. (e) Temperature time series from thermistors moored between 1 m and 20 m depth. Black circles indicate thermistor depths at each mooring deployment and white areas in indicate data gaps. Transparent gray regions mark the start and end of the surface melt season each year. Arrows in (d) and (e) indicate the timing of the mixing event shown in Fig. 8.



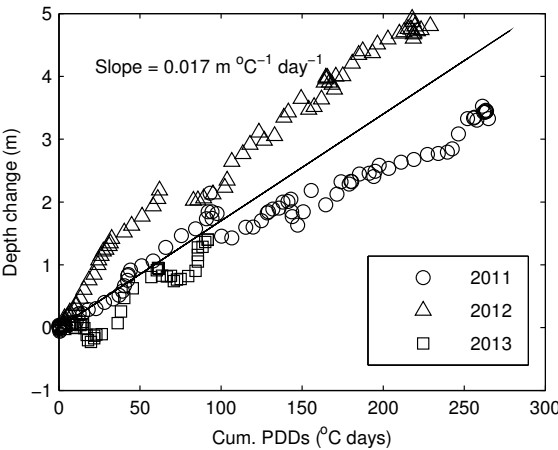

**Figure 7.** Correlation between the cumulative number of PDDs and the change in depth of the epishelf lake. Depth is estimated from the isotherm proxy during the melt season in 2011, 2012, and 2013.

Further evidence of the linkage between the magnitude of deepening of the halocline and surface conditions is provided by timeseries of CTD profiles collected during the field campaigns in 2012 and 2013 (Fig. 5). In 2012, the halocline deepened a total of 1.9 m over two months, from May 5th to July 9th. In 2013, however, over a similar two month period (May 10th to July 19th) the lake depth changed <0.5 m. Summer 2012 was unusually warm; a total of 93 PDDs had accumulated by the time the final profile was collected that year (09 July 2012). Summer 2013, however, was cool and a total of only 38 PDDs had accumulated by the time the final profile was collected that year (22 July 2013), despite the 2013 field campaign ending almost 2 weeks later than the 2012 campaign. None of the field campaigns, however, spanned the duration of the entire summer melt season, so the mooring data provides the most complete record of seasonal change. These observations indicate that freshwater inflow to the lake from surface meltwater runoff increased the depth of the lake each year, the amount of deepening directly proportional to the intensity of surface melting.

### 3.5.4 Mixing event in January 2012

An abrupt change in temperature, salinity, and depth of the halocline occurred on 11 January 2012 at 06:00 UTC (Fig. 6 and Fig. 8). Over a duration of 18 h the salinity at 13 m depth dropped from 22 to 12 g kg$^{-1}$ (Fig. 8a), and remained below 15 g kg$^{-1}$ for the remainder of the winter (Fig. 6d). At the same time, the heat content of the upper 25 m of the water column was relatively steady (apart from some fluctuations during the actual event), the slow rate of heat loss was not substantially different from the long-term average over winter (Fig. 8b). During the event the upper portion of the thermocline (above 11 m depth) was displaced upwards 1.5 m, while isotherms in the lower portion of the thermocline (below 11 m depth) spread apart vertically (Fig. 8c). High amplitude vertical fluctuations of isotherms persisted for several days following the event and were recorded from the uppermost thermistor (at 2 m depth) down to at least 50 m depth (not shown).



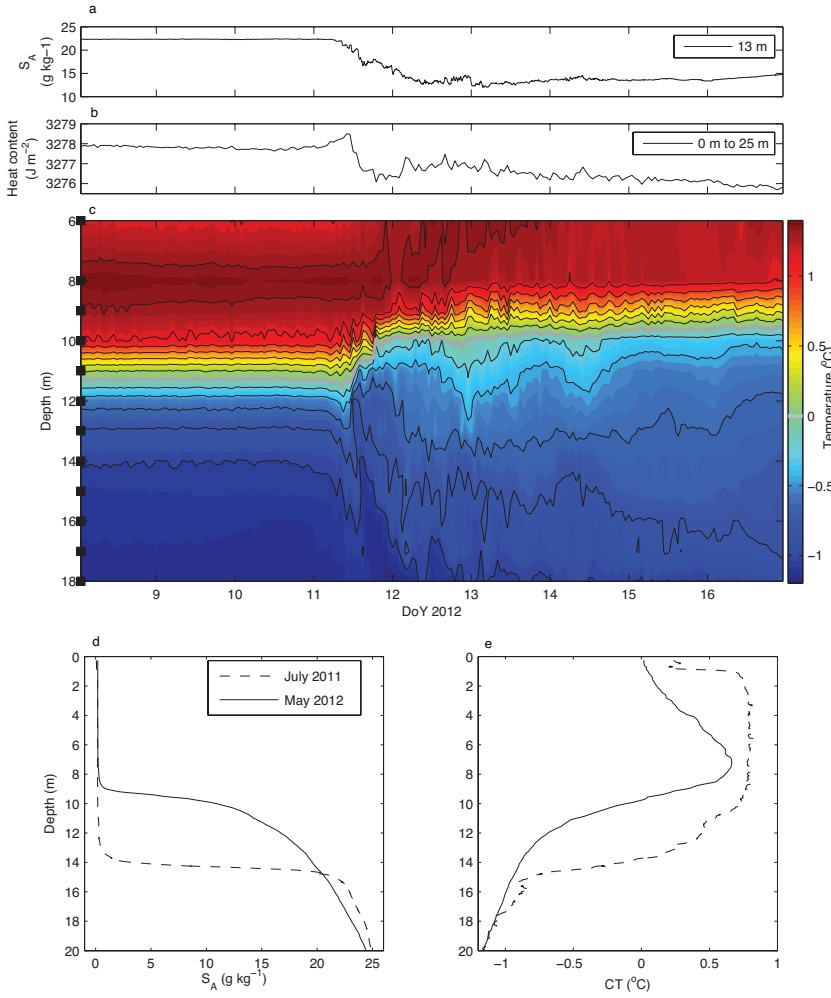

**Figure 8.** January 2012 MEL halocline mixing event. Time series of (a) salinity at 13 m depth, and (b) heat content between 0 m and 25 m, (c) temperature between 6 m and 18 m depth. The isotherm increment is 0.2°C and the 0°C isotherm (gray) is highlighted as a proxy for the bottom of the epishelf lake. Black squares on y-axis indicate the depths of thermistors. Profiles of (d) salinity and (e) temperature from field campaigns 6 months before (July 2011) and 4 months after (May 2012) the mixing event show the permanent change in the structure of the halocline (note that some of the change in the depth of the halocline is due to seasonal processes.





Profiles collected several months before and after the event showed a marked change in the depth of the epishelf lake and

the gradient of the lower thermo/halocline (Fig. 8 d and e). The changes were widespread and observed at several locations profiled throughout the fjord. These observations indicated that a sudden mixing event occurred at the bottom of the epishelf lake that entrained warm, relatively fresh water from the lake downward, and cool, salty water upward. The result was an abrupt, irreversible, thinning of the entire lake by 1.5 m.

### 3.5.5   Interannual

The complete record of lake depth inferred from CTD profiling between 1983 and 2014, as well as from the continuous mooring timeseries of temperature and salinity between 2011 and 2014 (Fig. 9), shows the depth of the lake decreased by a

total of almost 10 m over this period, yet there was substantial seasonal variation. Lake depths derived from both the moored salinity and temperature records match, and correspond to depths derived from CTD observations, providing confidence in the continuous multi-year isotherm proxy. The large degree of seasonal variation recorded by the mooring from 2011 to 2014 suggests that the depths of the lake determined from interannual CTD profiling prior to 2011 could have been influenced by the timing of profiling relative to the melt season. We have previously shown that the changes in depth of the lake during the

summers of 2011, 2012, and 2013, were correlated to the cumulated PDDs that season. Therefore, assuming the relationship between lake depth and PDDs was constant over the full record, and because all profiles were collected just prior to or during the meltwater inflow period, we adjusted the observed lakes depths ($z(N_{max}^2)$) to account for differences in the timing of observations and the intensity of the melt season using the accumulated number of PDDs at the time of profiling. We calculated a PDD-corrected depth ($z_{corr}$) by:

$$z_{corr} = z(N_{max}^2) - PDD(t)\frac{\Delta z}{PDD},\tag{4}$$

where $PDD(t)$ is the number of PDDs accumulated from the start of the melt season until the time of profiling, and $\Delta z/PDD$ is the ratio of seasonal changes in lake depth ($\Delta z$) to PDDs, taken as 0.017 m °C$^{-1}$ day$^{-1}$ (Fig. 7). The PDD-corrected depth was interpreted as the theoretical minimum depth of the lake prior to the onset of meltwater inflow each year.

The PDD-corrected depths (triangles; Fig. 9; Table 1) suggested that, in contrast to the raw CTD observations, the lake actually thinned monotonically every year since 1983 (with the possible exception of 2013–2014 which is discussed further below), although the rate of thinning increased substantially over time. Between 1983 and 2004 the change in the depth of the

lake was small, a total thinning of 0.9 m at an average rate of 0.04 ± 1.2 m a$^{-1}$ (the large error due to uncertainty in the lake depth estimated from the 1983 water bottle samples). The lake then thinned steadily from 2004 to 2011 at an average rate of 0.50 ± 0.05 m a$^{-1}$ (n=8, R$^2$ = 0.95, p <0.001). After this period there was a more dramatic phase of thinning, with the lake shoaling a total of 4.1 m from 2011 to 2012, with 1.5 m of that loss occurring during the January 2012 mixing event. This was followed by a further loss of 1.5 m from 2012 to 2013, and then an increase in depth by 0.6 m from 2013 to 2014 (although

we note that the lake depth inferred from the continuous mooring record indicates the lake may have actually continued to thin between 2013 and 2014 and the PDD-corrected lake depth from CTD profiling may not have captured the actual minimum





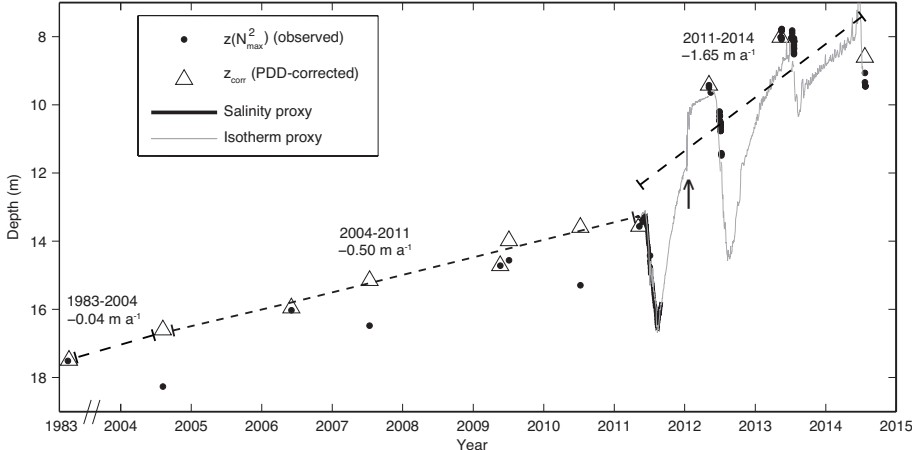

**Figure 9.** Interannual and seasonal changes in the depth of the MEL from 1983 to 2014. Shown are the observed lakes depths from CTD profiling ($z(N_{max}^2)$), PDD-corrected lake depths ($z_{corr}$; see Eq. 4) that account for the seasonal timing of the profile, as well as depths determined from moored salinity and temperature records from May 2011 to July 2014. Note the discontinuous x-axis between 1983 and 2004. To reduce clutter, $z_{corr}$ are shown only for the first profile collected during each field campaign after 2011. The arrow indicates the mixing event in January 2012. Dashed lines indicate a linear regression of $z_{corr}$ for the periods 1983–2004, 2004–2011 and 2011–2014.

depth in 2014). The average rate of thinning for the period 2011–2014 was $1.65 \pm 0.74$ m a$^{-1}$ (n = 4, R$^2$ = 0.71, p = 0.16), over three times higher than 2004–2011.

## 4 Discussion

Using a combination of data from CTD profiles, moorings, and remote sensing, we have documented changes in the water column structure, depth and extent of the MEL since 1983. Our results have shown that the lake has existed in close to its current spatial extent for the last three decades, and that the freshwater layer thinned substantially over the past decade. Our mooring records have shown that epishelf lakes are dynamic physical environments, undergoing depth changes of several metres in a single year owing, in part, to variation of meltwater inflow. In addition, epishelf lakes can be subject to abrupt

events that can dramatically shift the halocline over very short (<24 h) time periods. These results indicate that the utility of using changes in the depth of an epishelf lake to infer changes in the structure of the impounding ice shelf is dependent on an understanding of the freshwater budget and hydraulics of each system. In the following sections we first discuss possible causes of the January 2012 event given the magnitude of its effect on the depth of the halocline. Then we discuss the freshwater budget of the epishelf lake and what we can infer from our observations about changes in the thickness of the MIS and the

implications of these findings for other related systems.



### 4.1 January 2012 event

The catastrophic drainage of the Disraeli Fiord epishelf lake due to the fracturing of the WHIS (Mueller et al., 2003) provides a motivation to consider the rapid shoaling of the MEL in January 2012 as a sudden drainage of the lake due to a fracturing of the MIS. However, first-order calculations indicate that this was not likely the case. Assuming that the 1.5 m depth change observed in the MEL was due to drainage of an equal amount of epishelf lake water under the MIS, then the total volume change across the ~64.4 km² lake was 9.8 x 10⁷ m³. For all of this water to drain out of the fjord through the ~10 m wide basal channel in the MIS over the observed 18 h event duration, would have required a volume flux of 1.5 x 10³ m³ s⁻¹, requiring outflow velocities of >100 m s⁻¹. Even if the channel were an order of magnitude wider, the required velocities are still unrealistic, and we therefore consider it unlikely that the January 2012 event was related to a rapid drainage event.

In addition to shoaling of the thermocline (and by inference the halocline), the January 2012 event thickened the bottom half (i.e. water colder than ~0°C) of the thermocline (Fig. 8), suggestive of mixing processes. A scale for the energy required to cause such mixing is given by the change in background potential energy of the water column through adiabatic redistribution of density (Winters et al., 1995). Lacking salinity profiles close to the timing of the event, we idealize the system as two layers, with warm freshwater above and cool saltwater below, and utilize the moored temperature timeseries to estimate the portion of the water column that mixed across the interface, represented by the change in depth of the thermocline. Assuming the mooring is representative of the entire epishelf lake and that complete mixing occurs, the change in background potential energy ($\Delta PE$) is:

$$\Delta PE = \frac{1}{2} h_1 h_2 (\rho_2 - \rho_1) g A_L \tag{5}$$

where $h_1$ is the thickness of the portion of the upper layer involved in mixing, $h_2$ is the thickness of the portion of the lower layer involved in mixing, $\rho_1$ is the density of the upper layer (1000 kg m⁻³), $\rho_2$ is the density of the lower layer (1025 kg m⁻³), and $g$ is gravitational acceleration (9.81 m s⁻²) and $A_L$ is 65 km². From the temperature timeseries we chose a range of values for $h_1$ (0.5 m and 1.5 m) and $h_2$ (2 m and 7 m), which gives $\Delta PE$ of the order 10⁸ to 10¹⁰ J, which we consider an upper bound due to the assumptions in the model.

We have considered several possible sources that could provide sufficient kinetic energy to induce mixing on this scale, including a tidal anomaly, tsunami, earthquake, iceberg capsize, and subglacial outburst flood, yet lack convincing evidence that could identify a particular trigger as the cause. For example, the water level record shows no anomaly at this time that could indicate a large tidal excursion or tsunami occurred. A review of seismographic records from Ellesmere Island (http://earthquaketrack.com/r/ellesmere-island-nunavut-canada/recent) did not reveal any substantial earthquakes in January 2012. The increased production of icebergs in Milne Fiord from breakup of the glacier tongue and ice shelf suggests that turbulent energy released from calving or capsize of an iceberg could have generated mixing. Application of a scaling calculation (Eq. 4 in Burton et al., 2012) suggests the capsize of an iceberg with dimensions on the order of 50 x 100 x 150 m, typical of Milne Fiord, could have generated sufficient energy for mixing, however a review of available satellite imagery does not provide any clear evidence of a substantial mid-winter capsizing event. Likewise, we could find no clear evidence of the drainage of a supraglacial lake that might have triggered a sudden subglacial outburst flood, such as those observed in West Greenland



by Kjeldsen et al. (2014). Although the cause of the mixing event remains uncertain, and we cannot rule out other mechanisms, such as the propagation of an offshore anomaly into the fjord, it seems plausible that the anomalously warm air temperatures in

the summer of 2011 and the associated intense surface melting may have played a role in precipitating the mixing event. What our mooring records do clearly show is that continuous observations are the preferred method for tracking changes in epishelf lake depth, and where lacking, the possibility of abrupt changes in epishelf lake depth that are unrelated to changes in ice shelf thickness must be considered when interpreting interannual observations.

## 4.2 Freshwater budget

The depth of an epishelf lake at a moment in time is determined by both seasonal and long-term factors, primarily the balance between inflow, outflow, the area of the lake, and the depth of the ice dam. To separate out the influence of changes in the depth of the ice dam from the other factors it is necessary to consider the lake's freshwater budget. To do so, we can simplify

an epishelf lake to an idealized form, by making the following assumptions: that the level of the ice dam is fixed in time; has a simple bathymetry with a horizontal bottom and vertical sidewalls; and that there is no mass or volume flux across the halocline or through the ice walls. From conservation of volume the seasonal change in depth over time (dz/dt) can be expressed as:

$$\frac{dz}{dt} = \frac{Q_{in} - Q_{out}}{A_L}, \tag{6}$$

where $Q_{in}$ is the volumetric rate of inflow (m$^3$ s$^{-1}$) from surface runoff, $Q_{out}$ is the volumetric rate of outflow (m$^3$ s$^{-1}$) below

the ice shelf, and $A_L$ is the surface area of the epishelf lake (m$^2$), which is assumed constant over a season. The rate of inflow will be dependent on the magnitude of surface runoff from the surrounding glacial catchment. The rate of outflow will depend on the internal hydraulics of freshwater drainage below the ice shelf. Interannual changes in the area of the epishelf lake will be determined by advective and thermodynamic changes in the position of the ice margins that make up the shoreline of the lake. Knowledge of each of these variables can then provide a means to isolate factors controlling depth changes that are, and

are not, related to changes in the thickness of the ice dam. In the next sections we address what is known about these variables in the Milne Fiord setting.

## 4.3 Area

Changes in lake area could influence both the long-term changes in the depth of the lake and the seasonal response to inflow. On an interannual basis, if the volume of freshwater in the lake was conserved, an advective change in lake area could theoretically

cause a change in depth (for example, an advance of the Milne Glacier terminus would reduce lake area and lead to an increase in lake depth). However, the largest contributor to the increase in area of the lake appears to have been the thermodynamic transformation of ice of the Inner Unit of the MIS to epishelf lake water through melting, a process that would have increased both the area and volume of the lake. This process would not have altered the depth of the lake because the melted ice was already in hydrostatic equilibrium. Therefore, the five fold increase in area of the lake between 1959 and 1988 through melting

of the ice shelf is not expected to have had a large influence on the interannual depth of the lake. The small 10% fluctuations





in lake area between 2004 and 2014 are also insufficient to account for the 50% reduction in lake depth over this period, again indicating that area changes were not an important factor in driving the long-term shoaling of the lake.

On a seasonal basis, however, assuming the volume (and rate) of summer inflow was the same every year between 1959 and 1988, the five-fold increase in the area of the lake would have resulted in a five-fold decrease in the magnitude of summer deepening because the inflowing meltwater would have been distributed over a larger area. From 2004 to 2014, the 10% interannual fluctuation in the area of the lake would have resulted in an equivalent 10% variation in the magnitude of summer deepening, again assuming fixed inflow. Summer melt conditions, however, do vary substantially year-to-year; the annual cumulative PDDs varied by three-fold between 2004 and 2014. This suggests that annual fluctuations in the volume (and rate) of meltwater inflow appears to be the primary factor determining the magnitude of seasonal deepening of the lake.

## 4.4 Inflows

We have shown that there is a positive correlation between the seasonal depth increase of the epishelf lake in summer and the cumulative number of PDDs, suggesting a direct relationship between freshwater inflow to the lake and surface melting of snow and ice which is largely driven by air temperature. If, as a starting point, we assume that the observed increase in depth of the lake each year accounts for all runoff entering the fjord at the surface, and neglect outflow, then using our estimated depth change rate of 0.017 m PDD$^{-1}$ and the observed cumulative PDDs each year (ranging from 92 and 278 PDDs during the Milne Fiord weather station record from 2009 to 2014 ), equates to seasonal depth changes varying between 1.6 and 4.7 m. Combined with the average area of the lake (65 km$^2$) suggests an seasonal increase in volume of 100–300 x 10$^6$ m$^3$ a$^{-1}$ over this period. Given that we have neglected outflow, this estimate is a lower bound on the change in volume of the lake, and thus inflow from the fjord catchment.

The primary source of inflow to the lake is runoff from snow and glacier melt, and we can compare the seasonal volume change of the epishelf lake with an estimate of the total meltwater runoff from the Milne Fiord catchment. Lenaerts et al. (2013) estimated the average annual runoff from the 146 000 km$^2$ glaciated area of the Canadian Arctic Archipelago (CAA) from 2000 to 2011 was approximately 106 Gt a$^{-1}$. Applying this estimate to the 1108 km$^2$ glaciated area of the Milne Fiord catchment, which accounts for approximately 70% of the total area of the catchment, suggests the annual meltwater inflow to Milne Fiord is at least 1.12 x 10$^9$ m$^3$ a$^{-1}$. If all of this meltwater flows into the 65 km$^2$ MEL at the surface, the freshwater layer could theoretically deepen by 17 m each summer. In actual fact though, an unknown portion of this meltwater inflow likely enters the fjord at across the grounding line at the bed of the Milne Glacier and does not contribute to the MEL (our observations suggest perhaps 10-28% of the total runoff enters at the surface). Regardless of the exact proportions of surface versus subsurface inflow, this estimate provides independent evidence that there is ample inflow into Milne Fiord each summer to account for the observed seasonal depth increase of the epishelf lake. Furthermore, meltwater runoff from the CAA increased by 54% between 1971-2000 and 2000-2011 (Lenaerts et al., 2013), indicating that the long-term shoaling of the MEL can not be explained by a decrease of the inflow term in the conservation of volume (Eq. 6).

We have argued that long-term changes in both inflow and the area of the lake are insufficient to account for the long-term shoaling of the lake. Therefore, understanding factors affecting the volumetric rate of outflow are key to understanding





long-term changes in the depth of the lake. Next, we investigate the seasonal shoaling of the lake to better understand outflow hydraulics and use that information to inform what factors are driving the long-term shoaling.

## 4.5 Outflow

The mooring data showed that after inflow ceased each year, the lake thinned over winter until the following melt season. The rate of the seasonal shoaling of the halocline was non-linear, changing more rapidly at first and then gradually tapering off until the change with time was minimal just prior to the following melt season. The pattern indicated that the rate of shoaling depended on the relative depth difference between the halocline and the ice dam, suggesting an internal hydraulically controlled flow. Ice penetrating radar ice thickness surveys collected between 2008 and 2015 (Mortimer et al., 2012; Hamilton, 2016) indicate that the outflow pathway under the MIS is likely along a narrow basal channel following the east-west re-healed fracture in the Outer Unit. The measured ice thickness at a few locations along the mid-line of the basal channel is 8–11 m, suggesting the ice at the apex of the channel could be acting as a hydraulic control on outflow, and thus determining the

5 rate of change in the depth of the epishelf lake over winter. In this section we model the epishelf lake outflow as an internal hydraulically controlled flow through a weir, and compare simulated changes in epishelf lake depth over winter to mooring observations.

An idealized schematic of epishelf lake outflow through a basal channel in the ice shelf is shown in Figure 10, represented as a simple two-layer system, with freshwater overlying seawater ($\Delta\rho = 25$ kg m$^{-3}$). The ice dam acts as a hydraulic control,

10 limiting two-way transport below the ice shelf. If the depth of the seawater layer is much greater than the depth of the freshwater layer, then the situation is analogous to single layer flow through an inverted weir, but here the horizontal pressure gradient is supplied by the density difference between freshwater and seawater. For simplicity, if we assume a rectangular channel geometry and that the dimensions of the channel are constant in time (i.e., no melting or accretion on the ice walls and the depth of the ice dam is fixed), the volumetric outflow discharge ($Q_{out}$; m$^3$ s$^{-1}$) can be estimated using a modified form of the Kindsvater–Carter rectangular weir equation (Kindsvater and Carter, 1959):

$$Q_{out} = \frac{2}{3}\sqrt{2g'}C_e b h^{\frac{3}{2}},\tag{7}$$

where $g'$ is reduced gravity ($g' = g(\Delta\rho/\rho)$), $C_e$ is an empirically derived discharge coefficient, $b$ is the effective width (m) of the outlet channel, and $h$ is the effective depth (m) of the lake below the ice dam. $b$ and $h$ account for the effects of viscosity and wall friction and will therefore be somewhat smaller than the actual physical dimensions.

Assuming vertical sidewalls to the lake, the change in volume of the epishelf lake over time ($dV/dt$) is:

$$\frac{dV}{dt} = A_L \frac{dh}{dt}.\tag{8}$$

During winter, inflow is assumed negligible, so the change in volume is equal to the volumetric outflow (i.e., $dV/dt = Q_d$). Equating Eq 7 and Eq. 8, solving for $dh/dt$, and integrating gives:

$$h(t) = \left(\frac{1}{2}at + \frac{1}{\sqrt{h_0}}\right)^{-2},\tag{9}$$




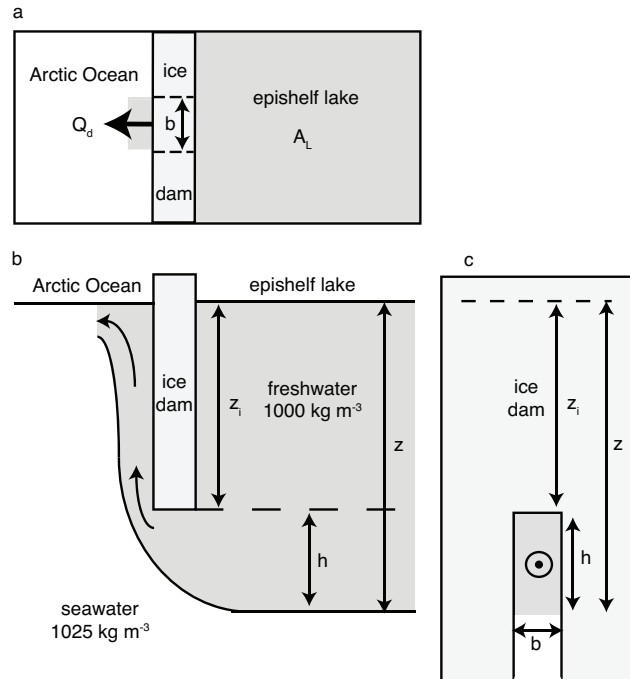

**Figure 10.** Schematic representation of epishelf lake outflow through a basal channel in the ice shelf dam in (a) plan, (b) elevation, and (c) cross-sectional views. The volumetric discharge is modelled using a modified form of the rectangular weir equation (Eq. 7).

where

$$a = \frac{\frac{2}{3}\sqrt{2g'}C_e b}{A_L}, \tag{10}$$

and $h_0$ is the initial depth of the lake below the ice dam at t = 0.

From Eq. 9 we modelled the change in depth of the epishelf lake for 275 days each winter of the three year mooring record, from approximately September to May of 2011–2012, 2012–2013, and 2013–2014. We assume, for the time being, that all changes to the depth of the ice dam occurred during the melt season (which was not modelled), so the depth of the ice dam remained fixed during each run (although differed between runs). The start time ($t = 0$) for each run was chosen as the date

when air temperatures permanently fell below 0°C for the winter, when meltwater inflow was assumed to cease. The initial depth of the lake ($z_{MEL}(t = 0)$) was estimated from the mooring temperature record. In the model, freshwater drains under the ice dam at a rate proportional to $h$ until $z_{MEL}(t)$ shoals to the level of the ice dam, $z_i$ (i.e., when $z_{MEL}(t) - z_i = h = 0$). The mooring records show that the lake was still shoaling when meltwater input commenced the following spring, suggesting the lake had not completely drained to the level of the ice dam. We therefore did not have a direct measure of $z_i$ which

was required to estimate $h_0$. Instead, we estimated $h_0$ as the sum of the difference between the initial lake depth and the depth when $t = 240$ elapsed days (the maximum duration of the shortest continuous mooring record each year), plus some





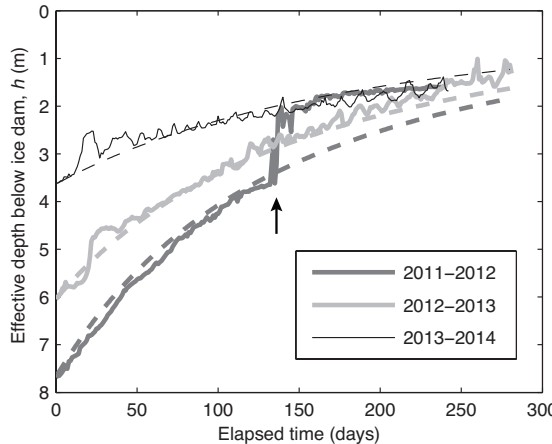

**Figure 11.** Change in the effective depth ($h$) of the epishelf lake below the ice dam over time during the winter of 2011–2012, 2012–2013, and 2013–2014. Elapsed time is measured from the end of the surface melt season each year. Observed depths (solid lines) are based on the isotherm proxy from the mooring record, while modelled depths (dashed lines) are based on a weir equation. Arrow indicates the mixing event that occurred in January 2012.

unknown offset $c_h$ (i.e., $h_0 = z_{MEL}(t=0) - z_{MEL}(t=240 \text{ days}) + c_h$). The actual depth of the ice dam at $t = 240$ days is then $z_i(t=240 \text{ days}) = z_{MEL}(t=0) - h_0$. Note that to account for the abrupt 1.5 m shoaling of the halocline in January 2012 we added an equivalent 1.5 m to $z_{MEL}(t=240 \text{ days})$ for that year. We found the best overall fit for all years was achieved

when $c_h = 1.6$ m and $C_e b = 4.5$ m, so these values were held constant for all runs.

    Modelled and observed changes to the depth of the lake relative to the ice dam are shown in Figure 11. Note that the depth of the ice dam differed for each run, so lake depths are plotted relative to the level of the ice dam for that year. Despite $h_0$ varying by over a factor of two among the different years (from 7.5 m in 2011–2012 to 3.3 m in 2013–2014), the simple drainage model simulated the observed pattern of changes in the depth of the lake each winter well. The model could not account for

the abrupt thinning of the lake in January 2012 (elapsed day 140 for 2011–2012) due to the mixing event, so observed and modelled values differ accordingly after this date. We note that the rate of shoaling of the lake slowed substantially after the mixing event, suggesting that the abrupt upward shift in the halocline moved it vertically closer to the level of the ice dam, thus reducing the rate of outflow. Overall, the results indicated that outflow drainage from the epishelf lake through the basal channel could generally be well simulated by weir outflow hydraulics.

5        To fully assess the model, we need to determine if the selected values for the parameters are physically realistic. For typical weirs the discharge coefficient, $C_e$, varies between 0.55 and 0.8 (ISO, 2008). If we assume that this range for $C_e$ is broadly appropriate for the MEL system, then the width of the channel, $b$, is between 5.6 and 8.2 m. This value is comparable to the estimated minimum width of the surface expression of the re-healed fracture at its narrowest point (2–8 m) from field observations and aerial imagery. Next, using the offset $c_h = 1.6$ m, we estimate the actual depth of the ice dam at the 240 day

10    mark, approximately May 1, varied from 9.4 m in 2012 to 7.5 m in 2014. This is broadly consistent with field measurements of



8 to 11 m thick ice measured along the re-healed fracture in July 2015. We conclude that the values chosen for the parameters are physically realistic and appropriate for this system, although further work is required to validate these.

The idealized model is useful in providing a simple explanation for the observed seasonal shoaling of the epishelf lake halocline over winter, and suggests that the geometry of the outflow channel is a key factor in determining the depth of the lake, on both a seasonal and interannual basis. Importantly, the only variable that changed between runs was $h_0$, which was ultimately dependent on the depth of the ice dam that year. For the modelled depth changes to match the observed differences in the rate of shoaling each year (i.e., fastest shoaling in 2011–2012, slowest in 2013–2014) required that the depth ice dam differed, getting shallower each year, suggesting the ice dam is indeed thinning over time. This, admittedly, highly simplified model, however, does not identify what mechanism led to the change in the level of the ice dam over time, such as basal melting (or accretion) along the channel, mass loss or gain on the surface or bottom of the ice shelf, and the hydrostatic adjustment of the ice shelf from these processes. In the following section we address what the long-term changes in the depth of the epishelf lake indicate about changes in the thickness of the ice shelf.

## 4.6 Changes in thickness of the MIS

Over the long-term, the depth of the epishelf lake is determined by the maximum depth of the ice along the lake's outflow pathway, referred to as the depth of the ice dam. Assuming the ice shelf is free-floating, shoaling of the ice dam could be caused by three mechanisms: 1) mean surface ablation thinning the ice shelf and causing an upward shift of ice dam due to hydrostatic adjustment; 2) localized submarine melting of the ice along the basal channel due to warm epishelf lake outflow leading to thinning the ice dam; or 3) submarine accretion of ice on the deeper portions of the ice shelf (not in the channel itself) thickening the ice shelf and causing an upward shift of the ice dam with hydrostatic adjustment (in this scenario lateral bridging stresses across the channel must allow ice in the channel to be raised out of hydrostatic equilibrium). Given that the Milne Ice Shelf is in a state of negative mass balance (Mortimer et al., 2012), the latter mechanism is unlikely, and the shoaling of the lake is likely due to either surface ablation or melting along the basal channel, or some combination of the two.

The long-term record of change in lake depth (Fig. 9), suggests that thickness of the ice dam was relatively stable from 1983 to 2004, but then thinned dramatically by almost 10 m between 2004 and 2014. From 2004 to 2011, changes in the depth of the lake suggest a steady thinning of the ice dam at a rate of $0.50 \pm 0.05$ m a$^{-1}$. After this period, there was a more rapid phase of thinning, with more interannual variability. Between May 2011 and May 2012 the lake abruptly shoaled by $4.1 \pm 0.4$ m, however, 1.5 m of this thinning was due to the mixing event in January 2012, suggesting the actual thinning of the ice dam over this period was $2.6 \pm 0.4$ m. From May 2012 to May 2013 the change in lake depth suggests the ice dam thinned a further $1.5 \pm 0.4$ m. Following these two years of rapid thinning, the rate of ice loss appears to have slowed, or even reversed (indicating thickening of the ice dam), between 2013 and 2014. The annual average rate of shoaling of the lake between 2011-2014 was 1.65 m a$^{-1}$, however, if we account for the 1.5 m shoaling due to the January 2012 event that did not appear related to a change in ice thickness, then this suggests the ice dam thinned at a rate of 1.15 m a$^{-1}$ over this period.

This pattern of accelerated ice loss is consistent with widespread rapid cryospheric loss along northern Ellesmere Island over the past decade. Since 2000, changes have included the break-up of the Ayles, Markham, Petersen, Serson, and Ward



Hunt ice shelves (Copland et al., 2007; Mueller et al., 2008; White et al., 2015a; Mueller et al., 2017a), the loss of multiyear

30  landfast sea ice (Pope et al., 2012), and the thinning of epishelf lakes along this coast (Veillette et al., 2008), including the long-term thinning and catastrophic drainage of the Disraeli Fiord epishelf lake (Veillette et al., 2008; Mueller et al., 2003). The accelerated thinning of the MIS after 2004 also corresponds to a period of rapid loss of perennial ice cover of Ward Hunt Lake, which lies 115 km to the east of Milne Fiord (Paquette et al., 2015), and a sharp increase in mass loss from glaciers and ice caps in the Canadian Arctic (Gardner et al., 2011; Lenaerts et al., 2013). These widespread changes strongly suggest that

35  regional climate warming is the driving factor behind the thinning of the MEL and MIS.

    Mortimer et al. (2012) showed that the MIS was in a state of negative mass balance between 1981 and 2009, thinning at an average annual rate of $0.29 \pm 0.1$ m $a^{-1}$ over this period, although with substantial spatial variability. Using average annual surface mass loss of $0.07$ m $a^{-1}$ recorded at the nearby WHIS between 1989 and 2003 (Braun, 2017), Mortimer et al. (2012) inferred that basal melting was a key contributor to overall thinning of the MIS over that period. However, focusing only on

a 7.5 km long repeat transect on the Outer Unit of the ice shelf that crossed the basal channel, Mortimer et al. (2012) found a lower average thinning rate of $0.08 \pm 0.08$ m $a^{-1}$ between 1981 and 2009. This rate of change is of the same order as that inferred from the epishelf lake records over a similar period ($0.1 \pm 1$ m $a^{-1}$ for 1983–2009). Thus, assuming surface ablation at MIS was equivalent to that at WHIS, then surface ablation alone appears sufficient to account for the thinning of the ice dam and the epishelf lake over the period 1983 to 2004. In fact, the lake depth observations suggest that the majority of ice dam

thinning between the 1980's and 2009 actually occurred between 2004 and 2009. Similarly, the majority of the surface mass loss between 1981 and 2009 on the WHIS actually occurred after 2002 (Braun, 2017)), observations which preceded an even more rapid phase of ice loss.

    Surface mass loss in Milne Fiord was recorded intermittently from 2009 to 2014 by a relatively sparse network of eight ablation stakes installed on the MIS and the Milne Glacier tongue, showing an annual area-averaged surface mass loss of 0.78

$\pm 0.64$ m $a^{-1}$ for this period (Hamilton, 2016). Accounting for hydrostatic adjustment of the floating ice shelf, this would have resulted in a shoaling of the ice dam by $0.69 \pm 0.56$ m $a^{-1}$, greater than the observed shoaling of the lake for 2004–2011, but less than that observed from 2011 to 2014. Thus, surface ablation alone appears sufficient to account for the change in depth of the epishelf lake prior to 2011, yet other mechanisms are required to explain the higher rate of thinning inferred from the epishelf lake records in 2011 and 2012.

We suggest that the rapid increase in mass loss inferred from the lake record after 2011 was fundamentally due to anomalously warm air temperatures in summer 2011 and 2012 which had a two-fold effect. First, the thinning of the MIS from enhanced surface ablation led to a upward shift of the ice dam through hydrostatic adjustment. Second, the increased surface melting resulted in a large seasonal meltwater inflow to the lake, substantial deepening of the halocline, and increased the outflow volume flux through the basal channel causing increased flux of heat to the ice walls and ceiling of the channel, ultimately

leading to enhanced melting and thinning of the ice dam. Therefore, although the long-term changes in the depth of the epishelf lake directly reflect the thickness of ice along the basal channel, they can not be used to infer a mean thinning rate of the entire ice shelf. The long-term change in the depth of the epishelf lake, however, provides a useful indicator of the state of the ice shelf, in particular the weakening of its structural integrity by thinning along the basal channel.



After the enhanced melting and shoaling of the ice dam during the warm summers of 2011 and 2012, the depth of the epishelf

lake remained relatively stable between 2013 and 2014. The summers of 2013 and 2014 were the coolest on record since the installation of the weather station in 2009, suggesting the significant reduction in surface ablation, meltwater production, and outflow through the basal channel combined meant that the ice dam did not substantially change in thickness over those years. Despite the short-term departure in 2013–2014 from the long-term thinning trend, however, it is likely that with continued regional climate warming the pattern of thinning of the lake and ice dam will continue. The warming trend continued in 2015 and 2016, with 2016 being the warmest summer on record at Milne Fiord (https://tinyurl.com/milnewx), and we expect thinning of the lake and ice dam was renewed.

### 4.7 Implications

Our results have shown that the timing of CTD profiling is critical to prevent aliasing the long-term, interannual record of

epishelf lake depth with seasonal changes. This is especially important if epishelf lake depth is to be used as a climate indicator, as suggested by Veillette et al. (2011b). In the absence of continuous records, the best long-term estimate for the lake depth is obtained from CTD profiling each year just prior to the initiation of the melt season, which for Milne Fiord would be June 1st. Observations at this time capture the annual minimum depth of the epishelf lake prior to inflow. Although the lake depth may not have reached equilibrium by early June, the depth measured at this time is arguably the most reliable indicator of the

long-term state of the lake, and likely the best indicator of the actual depth of the ice dam. Profiles collected during the melt season must account for the variations in summer meltwater inflow, and the method developed here based on PDD records provides an approach towards correcting for this variability.

The results from Milne Fiord have important implications for the interpretation of water column timeseries from other Arctic and Antarctic epishelf lakes. Veillette et al. (2008) observed interannual deepening of the epishelf lakes in Ayles and Markham

fjords, Ellesmere Island, on the order of 2 m between 2006 and 2007, however, the timing of data collection suggests that the change was due to seasonal variation of meltwater inflow, as subsequent profiles were collected later in the melt season. Similarly, Smith et al. (2006) suggested that interannual changes in the depth of the freshwater layer in two epishelf lakes at Ablation Point, Antarctica, between 1973 and 2001 could either indicate thinning of the George VI Ice Shelf, or alternatively, that the changes were due to seasonal processes. The 1973 observations from Smith et al. (2006) were taken in December,

over a month later in the austral summer than those collected in 2000/01, and we therefore suggest that seasonal processes were the more likely explanation. Continuous monitoring over a full seasonal cycle would, however, be required to verify this hypothesis.

Veillette et al. (2008) presented a 5-decade long record of CTD profiles from Disraeli Fiord that showed a change in the depth of the epishelf lake from 63 m in 1954, to 33 m in 1999, implying a 30 m change in thickness of the WHIS over this

period (note that we estimated this change from the level of the halocline in Veillette et al. (2008) Fig. 5a, not from the level of the 3 ppt isohaline as shown in Veillette et al. (2008) Fig. 5b). However, we caution that a shift in the timing of observations relative to the melt season (from September in 1954 and 1960, to June in 1999) means seasonal signals could have biased the long-term record. If we assume outfow of the Disraeli Fiord epishelf lake was restricted to a narrow basal channel in the WHIS




similar to that of the MIS, a likely scenario given the fracturing of the WHIS along a sinuous surface depression (Mueller et al.,
2003), then we can estimate the seasonal change in depth of the Disraeli Fiord epishelf lake from the volume of meltwater
inflow each summer. Lenaerts et al. (2013) estimated the average meltwater runoff from the CAA during the period 1971-2000
was 69 Gt $a^{-1}$, which if applied to the glaciated area of Disraeli Fiord (1385 km$^2$) suggests ~650 x 10$^6$ m$^3$ a$^{-1}$ meltwater
entered the fjord from glaciers (snowmelt from the 715 km$^2$ unglaciated area of the catchment only accounts for ~1 x 10$^6$ m$^3$
a$^{-1}$ assuming average annual snow accumulation of 0.15 m w. eq (Braun et al., 2004)). If all the runoff enters Disraeli Fiord
at the surface, this could have seasonally altered the depth of the 223 km$^2$ epishelf lake (Keys, 1978) by ~3 m. If we assume
maximal seasonal aliasing in the observations, then it is possible the actual change in depth was only 24 m, 20% less than that
inferred from the raw observations. Although the long-term record clearly indicates thinning of the WHIS along the outflow
pathway prior to breakup, this example highlights that accounting for seasonal effects is important if the rate of change of ice
shelf thickness is to be inferred from CTD profiles. The seasonal depth change will be most pronounced in epishelf lakes with
a small surface area, large seasonal inflow, and a narrow outflow channel.

Meltwater runoff from the CAA is predicted to more than double by the end of the 21st century (Lenaerts et al., 2013). In the
unlikely event that the MIS remains intact during this century, then seasonal changes in the depth of the MEL could double in
magnitude, further increasing the need to account for seasonal variation. However, given the long-term shoaling of the epishelf
lake, expanding network of fractures in the MIS, and predicted climate warming, further weakening of the ice shelf will likely
lead to a large scale calving event and complete drainage of the epishelf lake in the near future.

## 5    Conclusions

Our detailed observations from Milne Fiord show that the depth of epishelf lakes can vary substantially owing to seasonal
variations in meltwater inflow. This could potentially alias long-term records of lake depth from opportunistic CTD profiles
unless seasonal variations are properly accounted for, which requires continuous monitoring and knowledge of various factors,
including the inflow volume flux from the surrounding catchment, lake area and hypsometry, and outflow hydraulics under
the ice shelf. Accounting for these factors is particularly important if records of epishelf lake depth are used as proxies for
changes in ice shelf thickness. We emphasize even then that epishelf lake depth is only indicative of the minimum thickness
of the ice shelf along the outflow path of the epishelf lake, which for Milne Fiord appears to be a narrow basal channel in the
MIS. Monitoring ice thickness along a basal channel is important, however, as it provides a critical indication of the structural
integrity of the ice shelf. Observational verification of the outflow pathway would allow the basal channel hypothesis to be
confirmed or refuted, but more importantly, it would locate the thinnest and weakest fault line in the ice shelf and a likely site
for future large-scale fracturing.
The existence of epishelf lakes is highly sensitive to the interactions of the cryosphere with the atmosphere and hydrosphere.
The warming climate in the Arctic and Antarctic has resulted in an increased freshwater flux to the coast, with the potential to
increase the volume and depth of epishelf lakes. However, widespread climate-driven thinning and collapse of ice shelves is
outpacing the effects of increased freshwater inflow, leading to accelerated thinning and loss of these cryospheric systems in



the Arctic. At current rates of thinning the Milne Fiord epishelf lake, the last known epishelf lake in the Arctic, could be lost within the next decade, although continued breakup of the Milne Ice Shelf suggests a catastrophic drainage could occur at any time.

## 6 Code availability

Code used in analysis may be obtained from the corresponding author.

## 7 Data availability

Data may be obtained from the corresponding author.

*Author contributions.* We applied a sequence-determines-credit approach for the sequence of authors. AKH, BEL, and DRM conceived the study, AKH, DRM, WFV and LC conducted the field work, AKH and DRM analyzed the data, and AKH prepared the manuscript with contribution from all coauthors.

*Competing interests.* The authors declare that they have no conflict of interest.

*Acknowledgements.* This work was funded by grants from the Natural Sciences and Engineering Research Council (NSERC) of Canada, Canada Foundation for Innovation, Ontario Research Fund, University of Ottawa, and ArcticNet, a Network of Centres of Excellence of Canada. We thank the Polar Continental Shelf Program for providing excellent logistical support and Parks Canada for use of facilities. AKH was supported by graduate scholarships from NSERC and the Association of Canadian Universities for Northern Studies Garfield Weston Foundation, and awards from the Northern Scientific Training Program. We thank H. Melling, G. Lawrence, and R. Pieters for equipment loans and helpful advice on the research and an early version of the manuscript, and S. Brenner, K. Graves, J. Rajewicz, M. Richer–McCallum, D. Sarrazin, A. White, and N. Wilson for assistance in the field.





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



**Table 1.** Milne Fiord epishelf lake depth, area, volume and related observations.

| CTD Profiling Dates | CTD Profile Location | No. of CTD Profiles | MEL Mean Depth (±0.2 m) | PDDs to Date (°C days) (yr total) | MEL PDD Corr. Depth (±0.2 m) | Area (km²) | Volume (km³) | Image Source for Area Est. | Image Acquisition Date |
|---|---|---|---|---|---|---|---|---|---|
| 1959/08/17 | - | - | - | - | - | 13.5 | - | Aerial photo | 1959/08/17 |
| 1963/08/29 | - | - | - | - | - | 13.5 | - | Corona | 1963/08/29 |
| 1983/05/25 | PV | 1 | 17.5[a] | 0 (158)[*] | 17.5 | - | - | - | - |
| 1988/08/08 | - | - | - | - | - | 67.3 | - | SPOT-1 | 1988/08/08 |
| | | | | | | | | SPOT-1 | 1988/08/08 |
| 1992/01/29 | - | - | - | - | - | 60.6 | - | ERS-1 | 1992/01/29 |
| | | | | | | | | ERS-1 | 1992/03/16 |
| 1998/01/13 | - | - | - | - | - | 61 | - | RADARSAT-1 | 1998/01/13 |
| 2003/01/11 | - | - | - | - | - | 59.3 | - | RADARSAT-1 | 2003/01/11 |
| 2004/08/06 | NB | 1 | 18.3[b] | 98 (132)[*] | 16.6 | - | - | - | - |
| 2006/06/03 | NB | 1 | 16.0[b] | 4 (153)[*] | 15.9 | 65 | 1.04 | RADARSAT-1 | 2006/01/14 |
| 2007/07/13 | NB | 1 | 16.5[b] | 78 (215)[*] | 15.3 | - | - | - | - |
| 2009/05/29 - 05/30 | PV | 18 | 14.7 | 0 (274) | 14.7 | 65.2 | 0.96 | RADARSAT-2 | 2009/01/04 |
| 2009/07/04 | NB | 1 | 14.6 | 34 (274) | 14.0 | - | - | - | - |
| 2010/07/09 | NB | 1 | 15.3 | 100 (185) | 13.6 | - | - | - | - |
| 2011/05/10 | MM | 1 | 13.6 | 0 (278) | 13.6 | 67.6 | 0.92 | RADARSAT-2 | 2011/01/03 |
| | | | | | | | | RADARSAT-2 | 2011/02/28 |
| 2011/07/05 | NB | 1 | 14.4 | 94 (278) | 12.8 | - | - | - | - |
| 2012/05/05 - 05/14 | MM | 3 | 9.5 | 0 (253) | 9.5 | 64.4 | 0.61 | RADARSAT-2 | 2012/02/03 |
| | | | | | | | | RADARSAT-2 | 2012/04/17 |
| 2012/06/28 - 07/09 | MM, ML | 23 | 10.6 | 50 (253) | 9.8 | - | - | - | - |
| 2013/05/11 - 05/18 | MM, ML | 11 | 8.0 | 0 (92) | 8.0 | 67 | 0.54 | RADARSAT-2 | 2013/04/27 |
| 2013/07/04 - 07/22 | MM, ML | 46 | 8.1 | 10 (92) | 7.9 | - | - | - | - |
| 2014/07/12 - 07/24 | MM, ML | 13 | 9.3 | 39 (110) | 8.6 | 71.2 | 0.66 | RADARSAT-2 | 2015/03/27 |

PV - Purple Valley Bay

NB - Neige Bay

MM - Milne Fiord mooring

ML - multiple locations inside and outside Milne Fiord

[a]Jeffries (1985)

[b]Veillette et al. (2008)

[*]PDDs calculated from air temperatures interpolated from the Eureka weather station where $T_{Milne} = 0.87 * T_{Eureka} - 3.26$