# Peer review of "Dynamic response of an Arctic epishelf lake to seasonal and long-term forcing: implications for ice shelf thickness"

_The Cryosphere, 2017_

## Referee Comment (RC1) · Anonymous Referee #1 · 21 Apr 2017

Hamilton et al. present an extensive new dataset of hydrographic data from Milne Fiord epishelf lakes (MEL) in the Canadian Arctic combining both archive and extensive new CTD data. This is combined with ADCP, tidal height, mooring and AWS data to better understand the relationship between the depth of halocline and ice shelf thickness. Previous studies in the Arctic and Antarctic have used the depth of the halocline to infer long-term changes in ice shelf thickness. Whilst this simplistic model has been shown to hold true for some system, the overall complexity of this relationship has not been fully explored until now. I thoroughly enjoyed reading this paper - it is exceptionally thorough in its attempt to try and disentangle the various factors that control the depth of the halocline, involving the complex interaction of inflow and outflow of glacier melt,

lake area and depth. Whilst it is often difficult to combine legacy data – both in terms of comparing data from different instrument as well as highly pronounced seasonality – the authors have done all they can to ensure that the patterns they observe are robust. More generally the paper is very well written and the authors explore each of the factors that can influence lake depth/depth of the halocline methodically. Other than a few queries/comments below I don't have much to add and commend the authors on such a thorough job! Page 3, figure caption. 'will not penetrate to the freshwater layer because the vertical ascent of the buoyant plume.' Yes, but not always the case – especially if the Page 5, Figure 2. I'd include 'CTD locations' at top of white inset panel showing colour coded sites. Otherwise it's not immediately clear what they are. Page 6, Figure 3. Is this the only ice thickness data? A really nice thing to do in the future would be to measure ice shelf thinning/ice thickness and compare this with your detailed (and presumably ongoing?) hydrographic data. The ApRES system developed by the British Antarctic Survey/UCL would be ideal for this (Nicholls et al., A ground-based radar for measuring vertical strain rates and time-varying basal melt rates in ice sheets and shelves. Journal of Glaciology 61, 1079-1087). Also, label E-W fracture referred to on page 11, line 27. Page 6, section 2.2. Hydrography. Did you measure d18O of the lake water? I'm curious as to whether this would help refine your interpretation of the 2012 mixing event. As you say it is possible that this reflects a number of factors, although injection of a lot of glacier melt (possibly at depth) would be an interesting think to try and tease out. Page 11, section 3.3. Spatial extent. Is there any spatial bias in the seasonal changes observed between 2012 and 2013 plotted in Figure 5? If this is related to melt then spatial variability could be important? Page 22, line 19: Typo?! Likely enters the fiord at across the grounding line (remove 'at'?).

---

## Referee Comment (RC2) · Anonymous Referee #2 · 5 Jun 2017

Hamilton et al. present a novel investigation of the last known Arctic epishelf lake system, trapped between the Milne Fjord glacier tongue and proglacial ice shelf. The authors collected CTD, remote sensing and meteorological data on a range of timescales in order to determine what factors control seasonal and interannual lake extent and halocline depth. They then use the halocline depth record to reconstruct ice shelf thickness and state changes through time. The paper convincingly demonstrates a need for constraints on epishelf lake seasonality, hydraulics and fresh water budget in order to use halocline depth as a proxy for ice shelf thickness change. After taking these factors into account, the authors conclude that the lake and ice shelf are shoaling at an increasing rate. The lake could disappear within the next decade at the current

rate of ice shelf (ice dam) thinning.

In general, the paper is well written with precise, detailed and logical arguments. The authors present the best kind of process study – they describe the system in great detail with observations and then investigate the system's dynamics with simple, clear models and supporting math. In summary, I think the paper is well suited for publication in The Cryosphere.

However, I would like to raise several points that, if addressed, would strengthen the paper presentation and science. I hope these issues are straightforward – most are relatively minor. I follow these points with a small number of line edits.

First, the authors could perhaps be more careful in clarifying whether the epishelf lake depth is a proxy for mean ice shelf thickness or only the ice dam/basal channel area. Perhaps this difference is worth stating up front in the introduction? On a related note, the authors discuss (on page 27, second paragraph) the large spatial heterogeneity in ice shelf mass balance measured over the last few decades. The inner ice shelf (closest to the glacier) thinned an order of magnitude more than the outer shelf – potentially from differential submarine melt. However, the authors attribute lake shoaling to surface ablation, which would affect the ice shelf more homogeneously. Does this change the interpretation? Is most of the ice shelf responding to surface melt or basal melt?

Second, while the paper presents a succinct and convincing correspondence between seasonal lake (halocline) depth and PDD – the freshwater budget in section 4.2/4.4 would be strengthened by more explicitly considering inflow from submarine melting of the ice tongue, lake ice or ice shelf. Much of the ice tongue and ice shelf draft is above the halocline, which could result in large ambient melt fluxes, particularly as the lake warms in summer. It would be good to acknowledge/quantify this affect, or show that it does not have a significant role in setting lake stratification.

Third, section 4.4 notes that the majority (likely more than 70%) of meltwater runoff

enters the fjord as subglacial discharge. As freshwater, subglacial discharge will rise buoyantly to the halocline, possibly contributing to seasonal lake lowering and/or driving turbulent entrainment and submarine melting of the ice tongue. Can this process be more fully appreciated (mentioned) or discredited?

Fourth, as sort of an aside, I'm interested in understanding why the inner ice shelf (closest to the glacier) is the first to break up. This must be a consistent pattern in creating epishelf lakes. Through what mechanism might this occur? Is it due to glacier retreat?

Line comments:

Page 4, line 6 in 3rd paragraph: "an region" should be "a region"

Page 6, line 28: Move "We present this data" to the start of the sentence to make it more clear.

Section 3.1, first paragraph: It might make things smoother to state you'll investigate these factors with more detail in the sections that follow.

Section 4.1, third paragraph: Are large calving events and their oceanographic signature observable at other times in the record? It would be interesting to see.

Section 4.4, line 19: "across" and "at the bed" could be taken out in favor of "the fjord at the grounding line of the Milne Glacier..."

Figures: Figures 1, 2, 4, 6, and 8 could use larger label sizes similar to those in figures 3 and 5.

Figures 2 and 3 should have coordinates for reference.

---

## Author Comment (AC1) · 11 Jul 2017

Hamilton et al. present an extensive new dataset of hydrographic data from Milne Fiord epishelf lakes (MEL) in the Canadian Arctic combining both archive and extensive new CTD data. This is combined with ADCP, tidal height, mooring and AWS data to better understand the relationship between the depth of halocline and ice shelf thickness. Previous studies in the Arctic and Antarctic have used the depth of the halocline to infer long-term changes in ice shelf thickness. Whilst this simplistic model has been shown to hold true for some system, the overall complexity of this relationship has not been fully explored until now. I thoroughly enjoyed reading this paper - it is exceptionally thorough in its attempt to try and disentangle the various factors that control the depth of the halocline, involving the complex interaction of inflow and outflow of glacier melt, lake area and depth. Whilst it is often difficult to combine legacy data – both in terms of comparing data from different instrument as well as highly pronounced seasonality – the authors have done all they can to ensure that the patterns they observe are robust. More generally the paper is very well written and the authors explore each of the factors that can influence lake depth/depth of the halocline methodically. Other than a few queries/comments below I don't have much to add and commend the authors on such a thorough job!

We greatly appreciate the positive feedback on our manuscript and are very pleased the reviewer enjoyed reading the paper.

Page 3, figure caption. 'will not penetrate to the freshwater layer because the vertical ascent of the buoyant plume.' Yes, but not always the case – especially if the…

Unfortunately, it appears the reviewer's comment was cut off, however we interpret the comment to suggest that freshwater from subglacial discharge could indeed penetrate through the strong stratification of the epishelf lake halocline under some circumstances. For example, with a high volumetric subglacial discharge rate the plume could have sufficient upward momentum from buoyancy forces to overshoot the halocline before gravitational forces return it to its depth of neutral buoyancy. Under this situation it is possible that some subglacial freshwater could increase the volume and depth of the epishelf lake. However, observations from Hamilton (2016) appear to show that the subglacial discharge tends spread out in the lower halocline, below the stratification maximum, and thus does not contribute substantially to the freshwater thickness of the epishelf lake. We have revised the figure caption to reference these observations as follows (Page 3):

*Meltwater runoff discharged at the bed of the glacier entrains seawater on its ascent and observations suggest it spreads out below the epishelf lake halocline as a subsurface plume, not contributing to the thickness of the freshwater layer (Hamilton 2016).*

Page 5, Figure 2. I'd include 'CTD locations' at top of white inset panel showing colour coded sites. Otherwise it's not immediately clear what they are.

Figure legend revised as suggested.

Page 6, Figure 3. Is this the only ice thickness data? A really nice thing to do in the future would be to measure ice shelf thinning/ice thickness and compare this with your detailed (and presumably ongoing?) hydrographic data. The ApRES system developed by the British Antarctic Survey/UCL would be ideal for this (Nicholls et al., A ground-based radar for measuring vertical strain rates and time-varying basal melt rates in ice sheets and shelves. Journal of Glaciology 61, 1079-1087).

The ice thickness contour data shown is derived from numerous data sources, included ground-based IPR transects presented in Mortimer et al. (2012), more recently acquired ground-based ice-penetrating radar (IPR) transects, ICESat altimetry, and IceBridge aerial ice thickness and elevation. A description of the method for the development the ice thickness DEM is in Hamilton (2016). However, to answer the reviewer's question, yes, this is the only ice thickness data currently available, and supplementing it with timeseries data on ice shelf thinning would be ideal, but is outside of the scope of this publication. Regardless, we greatly appreciate the suggestion of the ApRES system, this is a valuable reference. We do have plans to install a similar autonomous radar system at the site in the future. Hydrographic monitoring of the site is ongoing, and we are focusing an upcoming project on melt rates in the basal channel.

Also, label E-W fracture referred to on page 11, line 27.

We have labelled the east-west fracture in both Fig. 2 (Page 5) and Fig. 3 (Page 6) for clarity.

Page 6, section 2.2. Hydrography. Did you measure d18O of the lake water? I'm curious as to whether this would help refine your interpretation of the 2012 mixing event. As you say it is possible that this reflects a number of factors, although injection of a lot of glacier melt (possibly at depth) would be an interesting think to try and tease out.

We appreciate the reviewer's excellent suggestion for addressing the cause of the mixing event using d18O data. We did collect several water samples in July 2012, six months after the mixing event, and analyzed the samples for d18O, among other constituents. However, we only obtained one sample (at 15 m depth) over the depth of the water column that was mixed (between approximately 10 and 25 m depth), thus the small sample size from within the mixed layer limits our ability to confidently compare its properties with the rest of the water column. As well, the addition of new meltwater during the intervening 6 months between the mixing event and the time of sampling changed the water column stratification so the water mass sampled had likely been altered by other mechanisms and was not representative of the mixed water. Given these factors, we do not feel our d18O data is robust enough for publication, nor could it further our ability to interpret the mixing event so we have therefore decided not to include discussion of the d18O dataset in the manuscript.

Page 11, section 3.3. Spatial extent. Is there any spatial bias in the seasonal changes observed between 2012 and 2013 plotted in Figure 5? If this is related to melt then spatial variability could be important?

The reviewer raises an interesting point, however, given the observations available we cannot discern any spatial bias in changes of the depth of the epishelf lake in 2012 and 2013. One might expect profiles near the grounding line to show a slightly deeper halocline if there was substantial runoff coming from the Milne Glacier, but we see no evidence for this. For example, in 2012 profiles were collected at numerous sites along the fjord on the same day (dark blue circles/profiles in Fig. 5), including near the grounding line, at the mooring site, and through the Central Unit and all showed equivalent halocline depth (within measurement error). We do see a substantially deeper halocline in profiles collected along the margins of the glacier tongue ten days later (red circles/profiles in Figure 5), however we do not have profiles from other locations along the fjord at that time to investigate spatial bias. We will definitely consider this in future sampling efforts and appreciate the reviewer's insight. We have added a sentence in this section to briefly discuss this as follows (Page 11, Lines 19-21):

*We observed no evidence of spatial bias in the data (above observational error) that could inform the distribution of meltwater sources or their temporal variation, although the sparse sampling may not have been sufficient to capture such variation.*

Page 22, line 19: Typo?! Likely enters the fiord at across the grounding line (remove 'at'?).

Changed as suggested.

Anonymous Referee #2
Hamilton et al. present a novel investigation of the last known Arctic epishelf lake system, trapped between the Milne Fjord glacier tongue and proglacial ice shelf. The authors collected CTD, remote sensing and meteorological data on a range of timescales in order to determine what factors control seasonal and interannual lake extent and halocline depth. They then use the halocline depth record to reconstruct ice shelf thickness and state changes through time. The paper convincingly demonstrates a need for constraints on epishelf lake seasonality, hydraulics and fresh water budget in order to use halocline depth as a proxy for ice shelf thickness change. After taking these factors into account, the authors conclude that the lake and ice shelf are shoaling at an increasing rate. The lake could disappear within the next decade at the current rate of ice shelf (ice dam) thinning.

In general, the paper is well written with precise, detailed and logical arguments. The authors present the best kind of process study – they describe the system in great detail with observations and then investigate the system's dynamics with simple, clear models and supporting math. In summary, I think the paper is well suited for publication in The Cryosphere.

We would like to thank Reviewer #2 for their supportive comments of our manuscript and appreciate their helpful comments for improving the paper that follow.

However, I would like to raise several points that, if addressed, would strengthen the paper presentation and science. I hope these issues are straightforward – most are relatively minor. I follow these points with a small number of line edits.

First, the authors could perhaps be more careful in clarifying whether the epishelf lake depth is a proxy for mean ice shelf thickness or only the ice dam/basal channel area. Perhaps this difference is worth stating up front in the introduction?

We agree with the reviewer's comment and have explicitly stated in the Introduction, and throughout the manuscript that epishelf lake depth is a proxy only for the ice dam thickness, not mean ice shelf thickness. We have edited the abstract as follows (Page 1, Line 3):

*Changes in the depth of the freshwater-seawater interface in epishelf lakes have been used to infer long-term changes in the minimum thickness of ice shelves, however, little is known about the dynamics of epishelf lakes and what other factors may influence their depth.*

And edited the introduction as follows (Page 2, Line 26):

*Measuring changes in the depth of the halocline separating freshwater and seawater could provide a relatively straightforward means to infer changes in the minimum thickness of the adjacent ice shelf (Vincent et al. 2001, Mueller et al. 2003, Veillette et al. 2008).*

We have also made minor changes throughout the manuscript to clarify that the focus is on minimum thickness of the ice shelf, not mean ice shelf thickness.

On a related note, the authors discuss (on page 27, second paragraph) the large spatial heterogeneity in ice shelf mass balance measured over the last few decades. The inner ice shelf (closest to the glacier) thinned an order of magnitude more than the outer shelf – potentially from differential submarine melt. However, the authors attribute lake shoaling to surface ablation, which would affect the ice shelf more homogeneously. Does this change the interpretation? Is most of the ice shelf responding to surface melt or basal melt?

First, we would like to directly answer the reviewer's questions: 1) we do not feel this changes the interpretation (and explain this further below), and 2) we expect the ice shelf is responding to both surface melt and basal melt, however the balance of these factors likely differs spatially and changes over time. The reviewer's question has helped clarify for us that discussion of this topic is improved by more explicitly stating that the epishelf lake depth is only a proxy for the minimum thickness of the ice shelf (i.e., the ice dam). Our best estimate for the location of the ice dam is along the east-west fracture in the Outer Unit of the ice shelf, thus changes in the epishelf lake depth will only reflect change in ice thickness along this channel (and perhaps the Outer Unit in general), but will not necessarily reflect thickness changes to the Central Unit (closer to the glacier).

Our interpretation of the results presented in Mortimer et al. (2012) and our own results is summarized in point form as follows:
- Surface ablation alone could explain the low rate of ice thinning of the Outer Unit of the Milne Ice Shelf between 1981-2009
- The shoaling of the epishelf lake over a similar period, from 1983-2009, is broadly consistent with the thinning rate of the Outer Unit.
- As per Mortimer et al. (2012), submarine melting is likely required to explain enhanced melting in other areas of the ice shelf, notably the Central Unit. We suggest that the elevated heat content of the epishelf lake is at least partially responsible for the higher submarine melt rate of the Central Unit (although differential surface ablation rates due to, for example, spatially variable wind-blown snow deposition and/or surface albedo could also play a role).
- After 2009 warming air temperatures increased surface ablation of the ice shelf. The rapid change in depth of the epishelf lake after 2009 can only partially be explained by ice shelf thinning due to this increased surface ablation, and we expect enhanced submarine melting also occurred.
- We suggest that the enhanced submarine melting was associated with an increase in the heat content to the epishelf lake due to higher rates of relatively warm meltwater inflow (and some contribution from solar radiation penetrating through the thin epishelf lake ice). The increased heat content of the lake would have led to higher submarine melt rate of the ice margins of the lake, including the inner portion of the Central Unit. In addition,

the increase seasonal outflow from the epishelf lake along the basal channel, may have also led to enhanced submarine melting of the ice dam, and thus interannual shoaling of the lake.

We have summarized the above points in Section 4.6 and we feel this does improve the clarity of the discussion (Page 27, Lines 17-31).

Second, while the paper presents a succinct and convincing correspondence between seasonal lake (halocline) depth and PDD – the freshwater budget in section 4.2/4.4 would be strengthened by more explicitly considering inflow from submarine melting of the ice tongue, lake ice or ice shelf. Much of the ice tongue and ice shelf draft is above the halocline, which could result in large ambient melt fluxes, particularly as the lake warms in summer. It would be good to acknowledge/quantify this affect, or show that it does not have a significant role in setting lake stratification.

We agree with the reviewer's comment that an additional source of freshwater to the epishelf lake is submarine melting of the ice shelf and glacier tongue above the halocline. As stated in the manuscript, we do not expect these sources to influence the depth of the halocline given that the ice is assumed to be in hydrostatic equilibrium. Regardless, it is useful to attempt to quantify the freshwater volume contribution to the lake from these sources for a comprehensive discussion of the freshwater budget and particularly with respect to the fact that heat from the epishelf lake may be a source for the thinning and breakup of the Central Unit of the ice shelf. We have added a new section to further discuss this topic (new Section 4.7; Page 29, Lines 1-26) as follows:

*The volume of freshwater input from submarine melt around the perimeter of the epishelf lake, as well as a submarine melt rate, can be constrained by considering a rough heat balance of the epishelf lake over winter. To obtain an upper bound on the melt rate, we simplify the heat balance by assuming all heat in the epishelf lake is lost to melting of ice around the lake perimeter. This assumes vertical heat flux through surface ice and heat flux across the halocline are negligible (and the latter avoids complications of advective heat loss from epishelf lake outflow below the ice dam). We further assume ice temperature is at its melting point. The change of mooring temperatures over each winter (dT), from August 15$^{th}$ to the following June 1$^{st}$ from 2011 to 2014, between 2 – 8 m depth varies from 0.5 to 2.5$^{o}$C. This equates to an annualized rate of heat loss (dH/dt) on the order of $10^{15}$ J a$^{-1}$ (where dH/dt = $\varrho_w c_p V_w$ dT/dt, where water density ($\varrho_w$) is 1000 kg m$^{-3}$, the specific heat capacity of water ($c_p$) is 4.18 x $10^3$ J kg$^{-1}$ $^{o}$C$^{-1}$, and the volume of the lake between 2-8 m depth is 3.75 x $10^8$ m$^3$). The rate of volume of freshwater input from submarine melting of perimeter ice ((dH/dt)/($L_i\varrho_i$)) is therefore on the order of $10^6$ to $10^7$ m$^3$ a$^{-1}$ (where the latent heat of fusion of ice ($L_i$) is 3.34 x $10^5$ J kg$^{-1}$ and density of ice ($\varrho_i$) is 900 kg m$^3$). We previously estimated that the annual volume of surface runoff into the lake was on the order of $10^8$ m$^3$ a-1 (i.e., 10-28% of the total meltwater runoff from the catchment). Thus, input from submarine melting around the lake's perimeter is a relatively small proportion (1-10%) of the total freshwater inflow to the lake and therefore has a relatively minor role in freshwater budget and stratification.*

*The volume of freshwater from submarine melting can be used to calculate a horizontal melt rate of the ice walls around the lake's perimeter using the surface area of the ice walls. The ice shelf*

*and glacier tongue form an ~40 km long ice perimeter to the lake, giving an ice wall surface area between 2 and 8 m depth on the order of $10^5$ $m^2$, resulting in an ice perimeter melt rate on the order of $10^1$ to $10^2$ m $a^{-1}$. We consider this melt rate an upper bound given the assumptions in the heat loss calculation and because the actual perimeter of the lake is substantially longer if the network of fractures in the MIS and MGT are taken into account. Although crude, this estimate does appear to be broadly consistent with the average annual increase in area of the epishelf lake observed by remote sensing between 2011 and 2014 (Table 1) of 1.2 ± 3.9 x $10^6$ $m^2$ $a^{-1}$, which if averaged around the ice perimeter of the lake suggests a melt rate on the order of $10^1$ ± $10^1$ m $a^{-1}$. Thus, submarine melting of the MIS and MGT driven by the relatively warm epishelf lake appears to be a plausible mechanism contributing to the areal expansion of the lake over the past few decades, and warrants further investigation.*

Third, section 4.4 notes that the majority (likely more than 70%) of meltwater runoff enters the fjord as subglacial discharge. As freshwater, subglacial discharge will rise buoyantly to the halocline, possibly contributing to seasonal lake lowering and/or driving turbulent entrainment and submarine melting of the ice tongue. Can this process be more fully appreciated (mentioned) or discredited?

Subglacial discharge of freshwater across the grounding line is certainly an important component of the freshwater budget of the fjord. However, water column observations collected during the summer melt season indicate the subglacial runoff plume appears to be largely confined to lower halocline, and has not been observed to penetrate above the halocline (Hamilton, 2016). Thus it does not appear that subglacial runoff contributes substantially to the depth of the epishelf lake, or the lowering of the halocline. The bulk salinity of the epishelf lake (<0.2 g kg$^{-1}$) suggests that while there is some small level of entrainment of seawater from below, the major inflow is very low salinity freshwater from surface snow and ice melt. In addition, water column profiles collected during summer show very little change in the stratification of the upper halocline, rather just a deepening of the halocline, indicating filling from above as opposed to upwelling of brackish subglacial discharge water from below. The Milne Glacier grounding line is at a depth of 150 m, thus any subglacial freshwater discharge will entrain substantial seawater on its ascent and appears from water column observations to find its level of neutral buoyancy at or below the depth of the stratification maximum in the epishelf lake halocline. We have added a sentence to address the influence (or lack thereof) of subglacial discharge on epishelf lake depth in Section 4.4 (Page 22, Lines 32-34) as follows:

*Water column profiles indicate that the subglacial freshwater plume entrains substantial seawater on its ascent and spreads out in the lower halocline (Hamilton 2016), likely not contributing substantially to the freshwater budget of the lake. Surface runoff therefore appears to be the main freshwater source to the epishelf lake.*

We also expect that subglacial discharge drives enhanced submarine melting along the base of the glacier tongue, however this process is beyond the scope of the present manuscript.

Fourth, as sort of an aside, I'm interested in understanding why the inner ice shelf (closest to the glacier) is the first to break up. This must be a consistent pattern in creating epishelf lakes.

Through what mechanism might this occur? Is it due to glacier retreat?

We agree with the reviewer that the pattern of ice shelf break up on the inner edge and the formation of the epishelf lake is an interesting phenomenon. The terminus of the Milne Glacier advanced 5.4 km between 1950 and 2015, so while it is possible that glacier retreat played a role in the initial formation of the epishelf lake between the glacier tongue and the ice shelf, it is not involved in the present pattern of the breakup of the ice shelf and epishelf lake expansion down the fjord. We suggest that the impoundment of surface runoff in the epishelf lake, with its elevated heat content, has led to enhanced submarine melting of the surrounding ice walls and thus preferential breakup of the inner margin of the ice shelf. We have included a paragraph at the end of Section 4.6 (Page 28, Lines 24-32) that discusses the epishelf lake's role in driving the breakup of the Central Unit of the Milne Ice Shelf as follows:

*Although the depth of the epishelf lake appears to be determined by changes in the thickness of the ice dam in the Outer Unit, the higher mean thinning rate of the Central Unit is an interesting phenomenon and warrants discussion. Unlike the Outer Unit, surface ablation alone does not appear sufficient to explain the high thinning rate of the Central Unit prior to 2009. While the spatial heterogeneity of surface ablation for the MIS is not well constrained, and differences in wind-blown snow deposition or surface albedo may be factors in the differential melt rates, it seems likely that the higher thinning rate of the Central Unit is due, in part, to enhanced submarine melting caused by the presence of the epishelf lake. The presence of the highly stratified epishelf lake results in an increased heat content within the fjord, and heat that is available year-round to drive submarine melting above the halocline. We therefore suggest that the presence of the relatively warm water of the epishelf lake is in fact contributing to the thinning and breakup of the Central Unit.*

Line comments:
Page 4, line 6 in 3rd paragraph: "an region" should be "a region"

Changed as suggested.

Page 6, line 28: Move "We present this data" to the start of the sentence to make it more clear.

Changed as suggested.

Section 3.1, first paragraph: It might make things smoother to state you'll investigate these factors with more detail in the sections that follow.

Changed as suggested.

Section 4.1, third paragraph: Are large calving events and their oceanographic signature observable at other times in the record? It would be interesting to see.

We appreciate the reviewer's suggestion related to exploring for evidence of other large calving events. Indeed, we have explored the entire 3+ year mooring timeseries for evidence of other mixing events to help explain the cause of the one observed in January 2012. Apart from

evidence of small episodic fluctuations in the halocline at other points in the record, which are not associated with significant mixing, there are no other events that compare with the magnitude of fluctuations or the extent of mixing observed in January 2012. There was some calving of the inner margin of the ice shelf in August 2012 when the lake ice of the epishelf lake partially broke up, however, we did not see any evidence in the mooring record of notable halocline variation or mixing at that time. Apart from a few capsized icebergs near the grounding line of the Milne Glacier, we have not observed many capsized icebergs in the fjord (those that calve from the Central Unit of the ice shelf are largely tabular in dimension and appear relatively stable year-to-year). We have included mention of the lack of field observations of capsized icebergs in the discussion as follows (Page 20 Lines 34-35; Page 21, Lines 1-2):

*Field observations of icebergs in the fjord up to 2015 indicate that most, particularly those calving from the ice shelf, are tabular and only a few glacier tongue-derived icebergs show evidence of capsize, perhaps explaining why halocline mixing events are rare (i.e., only one major mixing event observed during the multi-year mooring record).*

Section 4.4, line 19: "across" and "at the bed" could be taken out in favor of "the fjord at the grounding line of the Milne Glacier. . ."

Changed as suggested.

Figures: Figures 1, 2, 4, 6, and 8 could use larger label sizes similar to those in figures 3 and 5.

Changed as requested, although we note that some figures may be resized slightly in the final publication format and font appearance improved.

Figures 2 and 3 should have coordinates for reference.

Changed as requested.

Revised annotated manuscript follows.

[revised manuscript text omitted]